# Human glycogenins maintain glucose homeostasis by regulating glycogen metabolism

Tzu-Han Weng[1], Yu-Chung Pien [1,9], Ching-Jou Chen[1,9], Po-Pang Chen [2,9], Yu-Ting Tseng[2], Ying-Chen Chen[1], Wen-Po Hsiao[1], Ying-Ting Lee[1], Yi-An Chen[3], Yao-Chi Chen[4], Carmay Lim [4], Tzu-Han Hsu[2], Sung-Jan Lin [5,6,7], Hsin-Yung Yen [3], Kuo-Chiang Hsia [2] ✉ & Su-Yi Tsai [1,7,8] ✉

Proper regulation of glycogen metabolism is fundamental to cellular energy homeostasis, and its disruption is associated with various metabolic disorders, including glycogen storage diseases (GSDs) and potentially diabetes. Despite glycogen's role as an essential energy reservoir, the mechanisms governing its synthesis and structural diversity across tissues remain unclear. Here, we uncover the distinct physiological roles of the human glycogenins GYG1 and GYG2 in glycogen synthesis. Through cellular models, structural biology, and biochemical analyses, we demonstrate that, unlike GYG1, GYG2 exhibits minimal autoglycosylation activity and acts as a suppressor of glycogen formation. Together, these two glycogenins coordinate glycogen synthase activity and influence glycogen assembly in a cell-type-dependent manner. Importantly, these glycogenins modulate glucose metabolic pathways, thereby ensuring cellular glucose homeostasis. These findings address long-standing questions in glycogen metabolism and establish both GYG1 and GYG2 as critical regulators of glycogen synthesis and breakdown in human, providing insights with potential therapeutic implications for treating GSDs and metabolic diseases.

## Main

Glycogen, a highly branched polysaccharide, serves as a major energy reservoir in mammalian tissues[1]. The process of glycogenesis involves a series of enzymatic processes. First, glycogenin, encoded by *GYG*, generates an oligosaccharide primer through autoglycosylation[2,3]. Next, glycogen synthase (GS) elongates the primer by linking 10-12 glucose units via α−1,4 glucosidic bonds. Glycogen branching enzyme (GBE) then introduces branching points through α−1,6 glucosidic linkages, facilitating the formation of β glycogen particles with a diameter of 10 to 40 nm[4]. Further assembly of β glycogen particles with additional proteins leads to the formation of α particles in specific

[1]Department of Life Science, National Taiwan University, Taipei, Taiwan. [2]Institute of Molecular Biology, Academia Sinica, Taipei, Taiwan. [3]Institute of Biological Chemistry, Academia Sinica, Taipei, Taiwan. [4]Institute of Biomedical Sciences, Academia Sinica, Taipei, Taiwan. [5]Department of Biomedical Engineering, College of Medicine and College of Engineering, National Taiwan University, Taipei, Taiwan. [6]Department of Dermatology, National Taiwan University Hospital and College of Medicine, Taipei, Taiwan. [7]Research Center for Developmental Biology and Regenerative Medicine, National Taiwan University, Taipei, Taiwan. [8]Genome and Systems Biology Degree Program, National Taiwan University, Taipei, Taiwan. [9]These authors contributed equally: Yu-Chung Pien, Ching-Jou Chen, Po-Pang Chen. ✉e-mail: khsia@gate.sinica.edu.tw; suyitsai@ntu.edu.tw

tissues, such as the liver[4]. Notably, different tissues contain glycogen particles of distinct sizes that are tailored to their specific needs. For instance, β particles are predominantly found in the brain and skeletal muscle where there is a higher glycogen turnover rate[5]. In contrast, both α and β particles are found in the liver, where glycogen functions as energy storage[5]. However, how glycogen particle size and structure are regulated to adapt metabolic needs in different tissues remains largely unexplored.

Rodents, which only possess a single Gyg isoform, exhibit smaller β particles, with α particles being less prominent and inconsistently distributed across tissues[6]. However, primates, including humans, possess two distinct isoforms, GYG1 and GYG2[7,8]. GYG1 is expressed ubiquitously and is the sole isoform found in human skeletal muscle, whereas GYG2 is present in the liver, pancreas, adipose tissue, and heart[9], highlighting their tissue-specific distributions. Interestingly, smaller β particles are predominant in the brain and skeletal muscle, where GYG2 expression is very low or absent[10]. In contrast, larger α particles are abundant in hepatocytes, where GYG2 is highly expressed[10]. These observations imply a correlation between GYG isoform expression, glycogen particle size, and complex regulation of glycogen metabolism across tissues. Given the species- and tissue-specific expression patterns of GYG isoforms, investigating the distinct roles of GYG1 and GYG2 is crucial to understanding their contributions to glycogen metabolism in different tissues.

The overall protein domain structure of GYG is highly conserved, with an N-terminal Rossmann fold domain required for UDP-glucose binding and catalysis, as well as a C-terminal GS binding domain (Fig. 1a)[11,12]. In current models, GS and GYG1 form a heterooctamer composed of a GS tetramer and two GYG1 dimers, with each GS protomer binding to one GYG1 subunit[9]. Cryogenic electron microscopy (cryo-EM) structures of GS•GYG1 complexes have revealed that arginine phosphorylation in the N- and C-termini of GS inhibits its activity, inducing an inactive tense state[12]. This inactive state can be reversed by allosteric activation via G6P or dephosphorylation[12–14]. In addition to GS, GYG1 also modulates glycogenesis. GYG1 is an autoglycosylation enzyme that employs uridine diphosphoglucose (UDP-glucose) to synthesize a short glucose polymer attached to a tyrosine residue (Y195), thereby initiating glycogen synthesis. Residue Y195 serves as the autoglycosylation site in GYG1, with mutation of that site to phenylalanine (Y195F) having been shown to inhibit glycogenesis[12]. However, the interaction dynamics of GYG2 with GS and the functional significance of GS•GYG2 complexes remain largely unexplored.

Since GS, GYG, and GBE are the main enzymes that regulate glycogen synthesis, dysregulation of their enzymatic activities disrupts the homeostasis of glycogen metabolism, leading to glycogen storage diseases (GSDs) that are characterized by accumulations of excessive or abnormal glycogen, termed polyglucosan bodies (PB)[15]. Among them, mutations in GYG1 have been associated with muscle weakness and cardiomyopathy due to PB accumulation, leading to GSD type XV[16]. Notably, GYG1 mutations elicit distinct phenotypes across tissues: in skeletal muscle, they significantly reduce glycogen production, whereas in the heart they lead to an accumulation of PBs[16]. These findings indicate that GYG1 and GYG2 may coordinately play tissue-specific roles in regulating glycogen metabolism, with dysregulation contributing to pathologic phenotypes. Moreover, glycogen synthesis can proceed in the human liver even in the absence of GYG2[17], indicating a potential compensatory function of GYG1.

Understanding the differential contributions of GYG1 and GYG2 across various tissues is thus essential to unraveling the mechanisms underlying glycogen storage diseases and it could uncover new therapeutic targets. Notably, although studies of glycogenin-related disease have been carried out on mice, the presence of only a single glycogenin gene in this model organism limits its applicability to studying the functions of human glycogenins and their potential contributions to GSDs. Thus, we employed human embryonic stem cells (hESCs) as a model system to address these limitations. Furthermore, to understand the perturbation of glycogenesis in GYG-deficient patients, we integrated structural biology, biochemistry and metabolic analysis to elucidate the distinct roles of GYG1 and GYG2 in controlling glycogen synthesis and breakdown across different cell types, including skeletal muscle, neurons, hepatocytes, and cardiomyocytes (CMs).

Our study reveals that GYG1 and GYG2 exhibit dynamic expression patterns and distinct interactions with glycogen synthase (GS) that correlate with glycogen synthesis and breakdown. GYG1 consistently interacts with active GS under high glucose conditions, whereas GYG2 predominantly associates with the phosphorylated, inactive GS form during both glycogen synthesis and breakdown, a pattern of regulation that is disrupted in the absence of either GYG isoform. Through mechanistic analyses, we have found that the phosphorylated GS•GYG2 complex remains inactive and exhibits reduced autoglycosylation activity, leading to diminished GS activity and positioning GYG2 as a suppressor that negatively regulates glycogenesis. Notably, GYG2 controls glycogen particle size, promoting the formation of α particles, reflecting its specific role in adapting glycogen structure to varying energy demands in a tissue-specific manner. Our findings indicate that GYG1 supports glycogen synthesis under energy-abundant conditions and GYG2 modulates glycogen particle size to meet cellular energy requirements. Overall, our study demonstrates that the distinct roles of GYG1 and GYG2 are essential for fine-tuning glycogen metabolism to balance cellular energy needs.

## Results

### Both GYG1 and GYG2 regulate glycogenesis in hESCs

Given that GYG knockout mice exhibit glycogen over-accumulation across various tissues[18] and that glycogen synthesis can still occur in human liver without GYG2[17], GYG appears to be dispensable for glycogen synthesis. Since GYG2 levels are increased in some GYG1-deficient patients[19], first we wanted to examine if GYG2 may functionally compensate for GYG1. We examined the expression levels of GYG1 and GYG2 in hESCs by means of Western blotting (Fig. 1b), and employed the CRISPR/Cas9 technique to individually deplete expression of the respective genes from hESCs. Western blotting and DNA sequencing subsequently confirmed successful depletion of both isoforms (Supplementary Fig. 1a, b). Periodic Acid-Schiff (PAS) staining, which detects polysaccharides such as glycogen, revealed a significant reduction in glycogen synthesis in the GYG1 knockout (KO) hESCs compared to wild-type (WT) lines (Fig. 1c and Supplementary Fig. 1c). However, PAS staining of GYG2 KO hESCs showed comparable results to WT (Fig. 1d and Supplementary Fig. 1c). Notably, although glycogen levels were markedly reduced in the GYG1 KO hESCs, a few glycogen particles could still be stained and identified (Fig. 1c and Supplementary Fig. 1c).

Next, we examined protein levels of GYG1 and GYG2 in the GYG1 KO and GYG2 KO hESCs by means of Western blots and detected elevated levels of GYG2 protein in the GYG1 KO cells, but decreased levels of GYG1 in the GYG2 KO cells (Fig. 1e and Supplementary Fig. 1d, e (quantification results)). To unravel how GYG expression is potentially regulated, we generated "GYGs-free" double knockout (DKO) lines by ablating GYG2 in the GYG1 KO line (Fig. 1f and Supplementary Fig. 1f). PAS staining revealed even higher glycogen synthesis in these DKO lines than determined for GYG1 KO hESCs (Fig. 1g). Immunofluorescence staining using a glycogen antibody confirmed that the PAS-stained polysaccharides were indeed glycogen (Fig. 1g). Moreover, glycogen content analysis revealed a significant increase in glycogen content in both the GYG2 KO and DKO lines relative to WT, whereas it was reduced in the GYG1 KO line (Fig. 1h and Supplementary Fig. 1g). Although PAS staining of GYG2 KO cells showed minimal changes compared to WT (Fig. 1d), the quantitative glycogen assay (Fig. 1h) revealed a significant increase in glycogen

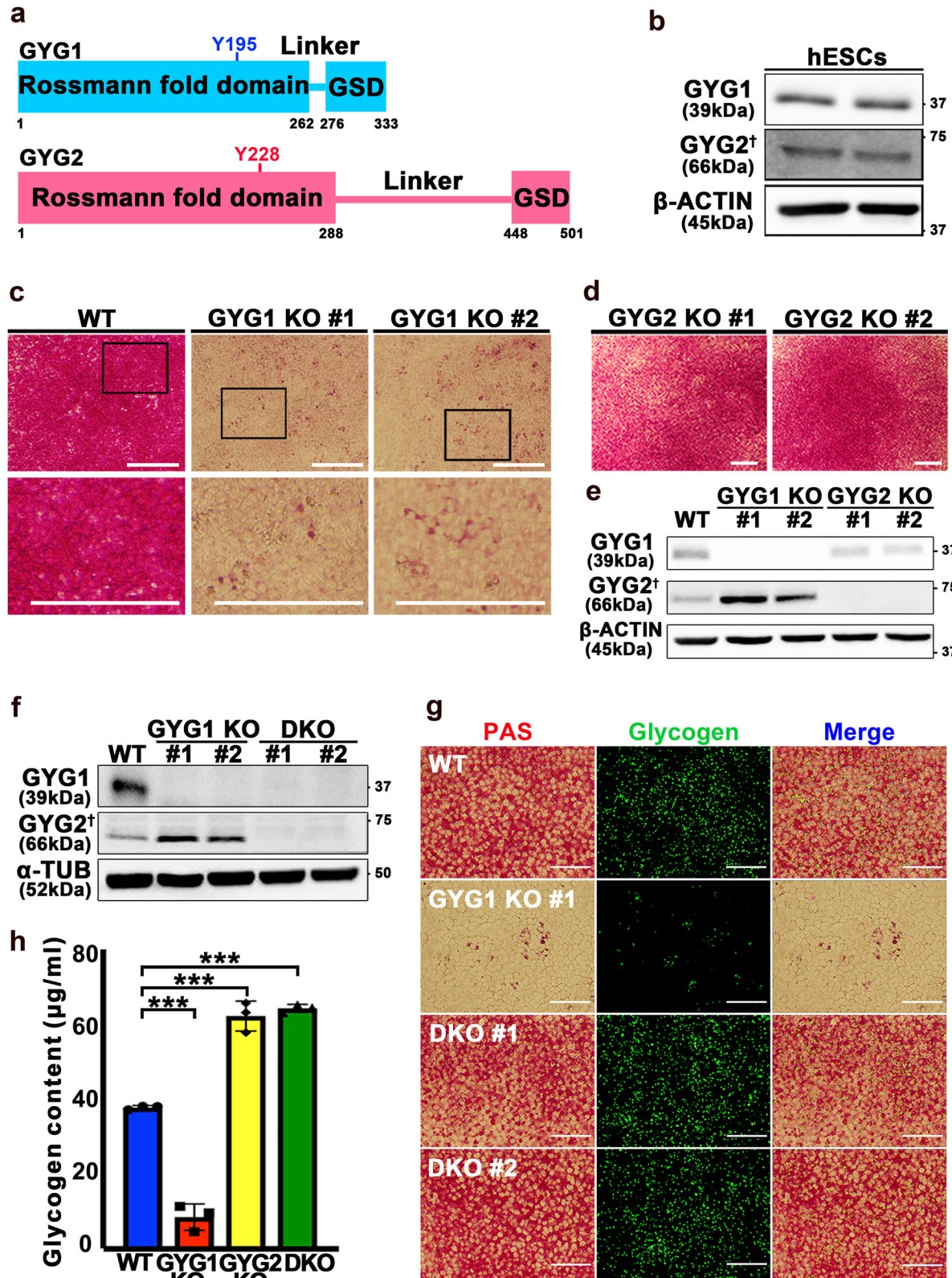

content. This discrepancy likely reflects the limited sensitivity of PAS staining in detecting modest changes in glycogen accumulation, particularly during early-stage metabolic alterations.

## GYG2 functions to suppress glycogen synthesis
We observed that (1) the level of GYG2 protein was increased in the GYG1 KO line (Fig. 1e), and (2) GYG2 KO and DKO exhibited elevated

glycogen content compared to the WT (Fig. 1h). Therefore, we hypothesized that GYG2 is unlikely to compensate for the functions of GYG1 in glycogen synthesis, but it might function as a suppressor to modulate glycogenesis. To test this hypothesis, we ectopically expressed GYG2 in both WT and DKO cell lines, referred to hereafter as WT^OE:GYG2 and DKO^OE:GYG2, respectively. Western blot analysis confirmed increased expression levels of GYG2 in both the WT^OE:GYG2

**Fig. 1 | Glycogen synthesis is reduced in GYG1 knockout hESCs. a** Domain organization of human GYG1 and GYG2 (numbers represent residue positions in the protein sequence). Auto-glycosylation sites (GYG1: Y195; GYG2: Y228) are indicated. GSD denotes the glycogen synthase binding domain. **b** Protein levels of GYG1 and GYG2 in hESCs, as detected by Western blot. β-Actin served as the loading control. **c, d** Representative Periodic acid-Sciff (PAS) staining images of GYG1 knockout (KO) (**c**) and GYG2 KO (**d**) hESCs. The wild type (WT) and two independent cell clones of each knockout line were examined. Scale bars: 200 μm. **e** Western blot analysis of GYG1 and GYG2 protein levels in GYG1 KO and GYG2 KO hESCs. β-Actin served as the loading control. **f** Protein levels of GYG1 and GYG2 in WT, GYG1 KO, and double knockout (DKO) hESCs, as detected by Western blot. α-Tubulin served

as the loading control. **g** Representative images of PAS (left panels) and immuno-fluorescence staining using glycogen antibody (middle panels) in the indicated cell lines, with overlay shown in the right panels (Merge). Two independent cell clones of the DKO line were examined. Glycogen is shown in green. Scale bars: 100 μm. **h** Bar graph of glycogen content in indicated cells. Glycogen contents were measured using a glycogen colorimetric kit. Data are mean ± standard deviation (SD) from three independent experiments. One-way ANOVA (Tukey's multiple comparison test); Statistical significance is indicated: ***$P < 0.001$. † Although the theoretical molecular weight of GYG2 is ~55 kDa (UniProt ID: O15488-1), the protein consistently migrates at ~66 kDa on SDS-PAGE, including when purified from insect cells (see Fig. 2i).

and DKO$^{OE:GYG2}$ lines (Fig. 2a). Notably, glycogen content in both the WT$^{OE:GYG2}$ and DKO$^{OE:GYG2}$ lines was significantly lower than WT (Fig. 2b, c), supporting the notion that GYG2 negatively regulates glycogen synthesis in hESCs.

## GYG2 binding modulates phosphorylation of GS
To explore how GYG2 regulates glycogen synthesis, we used Western blotting to examine levels of phosphorylated GS (pGS, i.e., the inactive form) in WT, GYG1 KO, GYG2 KO, DKO, and DKO$^{OE:GYG2}$ lines. We detected a significant increase in pGS levels from both the GYG1 KO and DKO$^{OE:GYG2}$ lines compared to WT (Fig. 2d, e). Moreover, knockout of the GS-binding domain (GSD) of GYG2 in the GYG1 KO line via a CRISPR/Cas9 approach elevated glycogen levels (Fig. 2f) and reduced pGS content (Fig. 2g). These results suggest an interaction between GYG2 and pGS in cell lysates, and imply that GYG2 binding to GS may alter GS phosphorylation states, leading to reduced glycogenesis.

Glycogen synthase (GS) exists in two isoforms: GS1 (*GYS1*), which is widely expressed in most tissues, and GS2 (*GYS2*), which is specific to the liver[20]. In this study, all GS-related experiments were conducted using GS1. Given that GYG2 binds GS, and their glycogen formation activity is associated with GS phosphorylation (Fig. 2f, g), we investigated if phosphorylation of GS influences the activity of the GS•GYG complex. To test this possibility, we treated GS•GYG complexes with lambda protein phosphatase (λPP) in an in vitro GS activity assay. We expressed human GS•GYG1, the GS•GYG1 (Y195F) mutant variant, and GS•GYG2 protein complexes in insect cells and then subjected them to chromatographic purification to isolate the proteins (Fig. 2h–j). In Western blots, recombinant WT GYG1 complexed with GS presented a smear (reflecting autoglycosylation), whereas the GYG1 (Y195F) mutant showed a single band (a lack of autoglycoslyation), consistent with findings reported previously[12] (Fig. 2h, i, k). Interestingly, GYG2 exhibited a single band in both SDS-PAGE and Western blots of GS•GYG2 complex (Fig. 2j, l), implying that it rarely undergoes autoglycosylation. When we treated GS•GYG1 and GS•GYG1 (Y195F) complexes with λPP (30 units), we observed a 2- and 5-fold increase, respectively, in GS activity relative to non-treated controls (Fig. 2m). Notably, even though the GS activity of the GS•GYG2 complex is relatively lower than that of GS•GYG1, λPP treatment of the GS•GYG2 complex enhanced GS activity ~4-fold (Fig. 2m), i.e., similar to the fold changes observed for the GS•GYG1 (WT) and GS•GYG1 (Y195F) complexes. These results indicate that phosphorylation may contribute to the suppression of GS activity in the GS•GYG2 complex. However, the persistently lower enzymatic activity, even after λPP or G6P treatment, indicates that GYG2 may regulate GS activity through additional mechanisms.

## GYG2 suppresses GS activity without inhibiting the G6P-GS interaction
In addition to the phosphorylation states of GS, G6P serves as a cofactor to regulate GS activity[19]. To investigate if GYG2 modulates G6P binding to the GS•GYG complexes, we performed native mass spectrometry on two complexes: GS•GYG1 (Y195F) and GS•GYG2. The

results reveal that, in addition to the signals corresponding to the GS•GYG2 or GS•GYG1 (Y195F) complexes in their *apo* states (Fig. 2n), multiple satellite peaks were observed for both complexes upon co-incubation with G6P at an unsaturated concentration (Supplementary Fig. 2a–c). Deconvolution of mass spectra into mass domains further indicated that mass differences of two species adjacent to *apo* complexes are close to G6P molecules, indicative of multiple G6P binding events (Fig. 2n). Notably, in the presence of 2 mM G6P, the activity of the GS•GYG1 (WT) and GS•GYG1 (Y195F) complexes increased significantly (by 2- and 4-fold, respectively) compared to controls without G6P treatment (Fig. 2m). Similarly, the GS•GYG2 complex showed a comparable ~5-fold increase in GS activity regardless of the presence or absence of G6P (Fig. 2m), suggesting that GYG2 binding does not inhibit the interaction of G6P with GS.

In addition, after successfully isolating GYG2 protein (Fig. 2l and Supplementary Fig. 2d), we further conducted a competition assay by introducing varying amounts of GYG2 proteins to the GS•GYG1 (WT) complex. This resulted in a concentration-dependent decline in GS activity (Fig. 2o), indicating that GYG2 alone can inhibit GS activity, even though this binding does not interfere with the binding of G6P to GS.

## The limited autoglycosylation activity of GYG2 inhibits GS activity
We have shown that the glycogenesis capability of the GS•GYG2 complex is comparable to that of GS•GYG1 (Y195F) (Fig. 2m). Therefore, we investigated how the low autoglycosylation activity of GYG2 impacts glycogen synthesis. In addition to Western blot analysis (Fig. 2l), we conducted PAS staining to further validate if GYG2 can undergo autoglycosylation as part of the GS•GYG2 complex. In native gels, PAS staining revealed polysaccharide signal for the GS•GYG1 (WT) complex but not for the GS•GYG1 (Y195F) or GS•GYG2 complexes (Fig. 3a, b), confirming that GYG2 cannot undergo autoglycosylation. Notably, examining the GS activity of these three complexes in parallel revealed 10- and 84-fold lower glycogen activity for the GS•GYG1 (Y195F) and GS•GYG2 complexes relative to GS•GYG1 (WT), respectively (0.30 ±0.06 and 0.03 ±0.005 versus 2.86 ±0.50; Fig. 3c), indicating that binding of "non-glycosylated GYG2" to GS suppresses glycogenesis. In this context, GYG2 functions as a non-productive binding partner that neither activates GS nor supports primer formation and may restrict GS activity by stabilizing a conformation that is less permissive to activation.

As noted above, GYS2 (GS2) is predominantly expressed in the liver and has been reported to interact with GYG2[20]. To account for this scenario, we also purified the GS2•GYG2 complex (Supplementary Fig. 2e, f) and performed PAS staining, which resulted in no detectable polysaccharide signal (Supplementary Fig. 2g), similar to what we observed for the GS1•GYG2 complex (Fig. 3a, g), indicating that GYG2 exhibits low autoglycosylation activity regardless of its binding partner. In addition, GS activity assays showed that the GS2•GYG2 complex displayed similarly low enzymatic activity to that of the GS1•GYG2 complex (Figs. 2m, 3h), irrespective of the presence of λPP or G6P

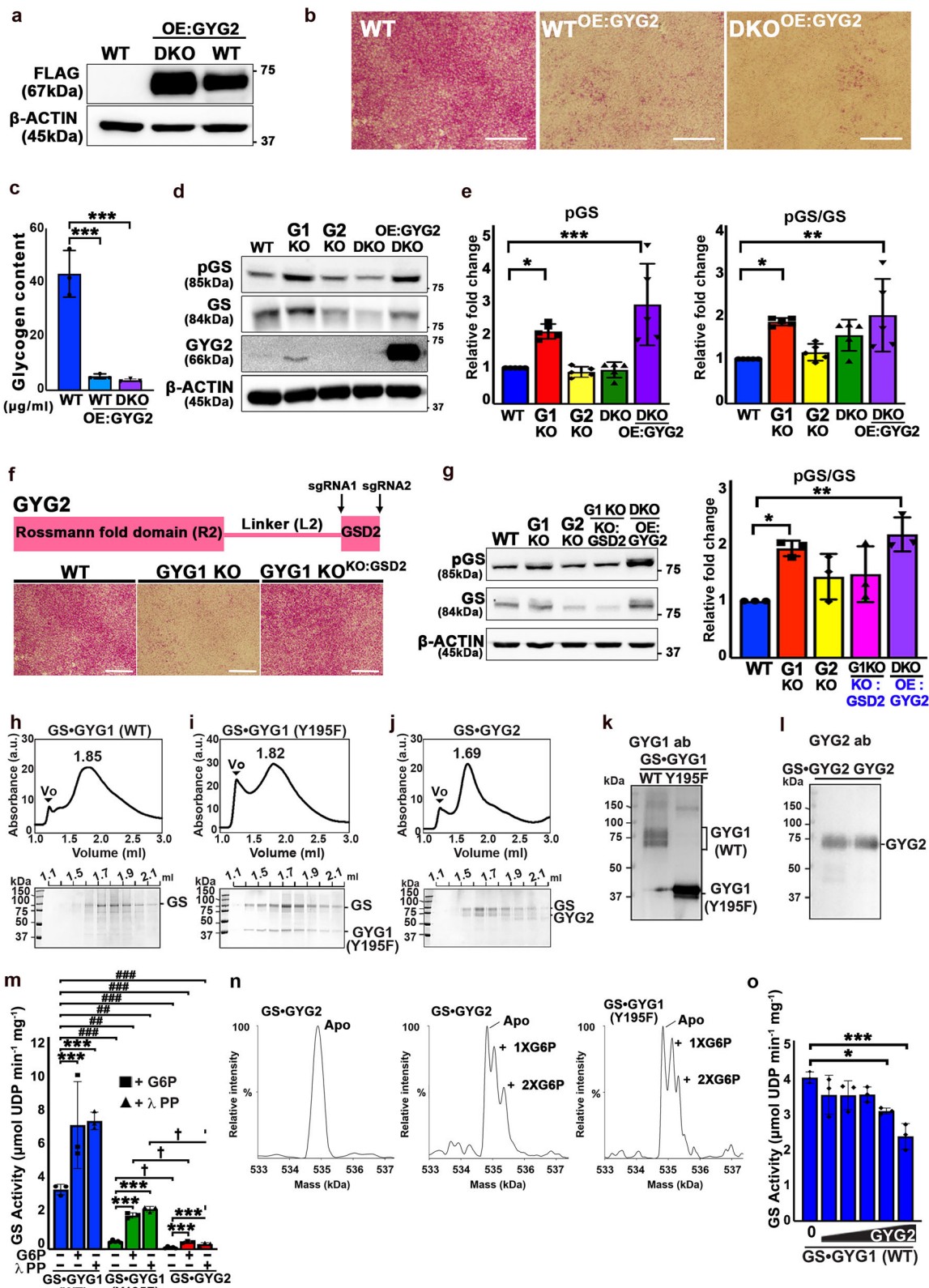

(Supplementary Fig. 2h). Together, these results demonstrate that GYG2 is unable to effectively initiate glycogen synthesis, regardless of its association with GS1 or GS2 (Supplementary Fig. 2e–h), reinforcing its role as a non-initiating glycogenin.

Next, we aimed to elucidate the significance of glycosylation on the Rossmann fold domain of GYGs in terms of GS activity by performing a domain-swapping experiment between the GYG isoforms GYG1 and GYG2. Therefore, we replaced the Rossmann fold domain of GYG2 (rarely autoglycosylated) with that of GYG1 (actively auto-glycosylated), but retained the GYG2 linker (L2) and glycogen synthase domain (GSD2), resulting in a chimeric GYG1 construct (Fig. 3d). Subsequently, we purified the GS•chimeric GYG1 complex to homogeneity and evaluated its autoglycosylation and GS activities (Fig. 3e, f). Western blot analysis using anti-GYG1 antibody revealed a smear pattern

**Fig. 2 | GYG2 negatively regulates glycogen synthesis. a** Western blot of ectopic GYG2-FLAG expression in WT and DKO hESCs using anti-FLAG ($n = 3$). **b** PAS staining of WT, WT$^{OE:GYG2}$, and DKO$^{OE:GYG2}$ hESCs. Scale bars: 200 μm. ($n = 3$). **c** Glycogen quantification using colorimetric assay in indicated cell lines. Data are mean±SD ($n = 3$). One-way ANOVA (Tukey's test); \*\*\*$P < 0.001$. **d** Western blot of pGS and GS in the indicated cell lines. **e** Bar graphs showing pGS and the ratio of pGS to GS. Data are mean ± SD ($n = 5$). One-way ANOVA (Tukey's test); \*$P < 0.05$, \*\*$P < 0.01$, and \*\*\*$P < 0.001$. **f** Domain organization of GYG2 with sgRNA sites. PAS staining of WT, GYG1 KO and GYG1 KO$^{KO:GSD2}$ hESCs ($n = 3$). Scale bars: 200 μm. **g** Western blot of pGS and GS (left) and quantification of pGS/GS (right). β-Actin was used as loading control. Data are mean±SD ($n = 3$). One-way ANOVA (Tukey's test); \*$P < 0.05$, \*\*$P < 0.01$. **h–j** Size exclusion chromatography (Superose 6) elution profiles for GS•GYG1 (WT) (**h**), GS•GYG1 (Y195F) (**i**), and GS•GYG2 (**j**). Void volumes (Vo) are indicated. SDS-PAGE with Coomassie staining of peak fractions (bottom).

$n = 3$. **k** SDS−PAGE and Western blot of WT/mutant GS•GYG1 complexes using anti-GYG1. **l** Western blot of GS•GYG2 and GYG2 alone using anti-GYG2. **m** GS activity of GS•GYG1, GS•GYG1 (Y195F), and GS•GYG2 with or without 2 mM G6P and 30 units lambda phosphatase (λPP). Asterisks \* represents statistical comparisons relative to controls of each complex with or without G6P or λPP. # represents statistical comparison of GS•GYG1 (Y195F) and GS•GYG2 with the GS•GYG1 control. † represents statistical comparison of GS•GYG1 (Y195F) with GS•GYG2 under G6P or λPP treatment. Data are mean ± SD ($n = 3$). Two-way ANOVA (Tukey's test); \*$P < 0.05$, \*\*$P < 0.01$, \*\*\*$P < 0.001$, # $<0.05$, ## $<0.01$, ### $<0.001$, †$P < 0.05$, ††$P < 0.01$, †††$P < 0.001$. **n** Native MS of GS•GYG2 and GS•GYG1 (Y195F) with G6P. Mass spectra show apo and G6P-bound states with binding stoichiometry. **o**, GS•GYG1 with increasing GYG2 before activity assay. Data are mean ± SD ($n = 3$). One-way ANOVA (Tukey's test); \*$P < 0.05$, \*\*\*$P < 0.001$.

for the GS•chimeric GYG1 complex (Fig. 3f), akin to that detected for the GS•GYG1 (WT) complex (Fig. 2h, bottom panel), and PAS staining revealed positive signal, as also determined for the GS•GYG1 (WT) complex (Fig. 3g). Importantly, the GS•chimeric GYG1 complex exhibited much greater GS activity (40-fold) than the GS•GYG2 complex, i.e., comparable to the GS•GYG1 (WT) complex (Fig. 3h). Furthermore, fold changes in GS activity for all complexes—GS•GYG1 (WT), GS•GYG2 and GS•chimeric GYG1—were comparable in the presence or absence of λPP or G6P (Fig. 3h). Together, these results indicate that glycogenin autoglycosylation likely plays a critical role in regulating glycogenesis in complex with GS. To further validate these biochemical findings, we cloned GYG1, GYG2, and chimeric GYG1 into a lentiviral vector and introduced these viruses into the DKO line (Fig. 3i). Consistently, a glycogen content assay demonstrated that glycogen levels in cells infected with the chimeric GYG1-hosting virus were similar to those in DKO$^{OE:GYG1}$ cells (Fig. 3j), thereby corroborating our biochemical findings that GYG autoglycosylation contributes significantly to glycogenesis.

Thus, glycogen synthesis can be regulated by the Rossmann fold domain. To further investigate this notion, we generated a Y228F mutant of GYG2 by substituting the putative autoglycosylation site tyrosine with phenylalanine to test its specificity. After purifying the GS•GYG2(Y228F) complex (Supplementary Fig. 2i, j), we assessed its activity by means of PAS staining and GS activity assays. The GS•GYG2(Y228F) complex not only presented no PAS staining signal, but also exhibited similarly low activity to the wild-type GS•GYG2 complex (Supplementary Fig. 2k, l). These results reveal that residue Y228 in GYG2 does not mediate the autoglycosylation activity necessary for glycogen synthesis. Thus, restoring this activity would require reprogramming of the entire Rossmann fold domain.

### The flexibility of Y228 limits autoglycosylation of GYG2

To gain molecular insights into how GYG2 regulates GS activity, we used cryo-EM to determine the structure of the GS•GYG2 complex. First, peak fractions from size exclusion chromatography were vitrified on cryo-EM grids and then micrographs were acquired using a Titan Krios microscope (300 keV) (Supplementary Fig. 3a). Two-dimensional (2D) class averaging revealed secondary structural features of the GS•GYG2 complex (Supplementary Fig. 3b). Next, multiple rounds of three-dimensional (3D) reconstruction were performed using the best-resolved classes across each classification (Supplementary Fig. 3c). Ultimately, we determined a cryo-EM structure of the GS•GYG2 complex, representing a heterooctameric protein complex with an average resolution of 2.84 Å from 238,004 particles (Supplementary Fig. 4a–d and Supplementary Table 1).

We were able to assign the orientations of GS structures in the resulting cryo-EM densities based on features defined by previously-generated cryo-EM structures of GS•GYG1 (PDBs 7Q0B and 7Q12; Fig. 4a). In particular, both GYG1 (i.e., GSD1) and GYG2 (i.e., GSD2) use their C-terminal GS-binding domains to interact with GS, with the

additional density lying above GS in our GS•GYG2 cryo-EM structure being attributed to the GSD of GYG2 (Fig. 4a). Root mean square deviations (rmsd) of 1.037 and 9.469 Å (Cα; as assessed in Pymol) between GS•GYG2 and GS•GYG1 in the inhibited or active state, respectively, underscore the minimal differences between the GS•GYG2 complex structure and that of GS•GYG1 in the inhibited state (Fig. 4b, c). Our structural information also indicates that the GS•GYG core structure is comparable between the GS•GYG2 and GS•GYG1 complexes, implying they should share common dephosphorylation and G6P activation regulatory mechanisms.

To understand why the catalytic Rossmann fold domain of GYG2 elicits lower autoglycosylation activity, leading to diminished glycogenesis, we compared primary protein sequences and AlphaFold3-predicted structures of GYG1 and GYG2 across different species (Fig. 4d–f; Supplementary Fig. 4e). The predicted catalytic Rossmann fold domain structures of GYG1 and GYG2 across different species showed minimal rmsd values (0.132 to 0.23 Å in GYG1 and 0.394 to 0.532 in GYG2), indicating conservation among the predicted structures (Fig. 4d). A previous study proposed that GYG1 undergoes conformational rearrangements in three key regions that modulate its activity[11], including the lid segment, the acceptor arm containing the Y195 acceptor residue, and the C loop adjacent to the acceptor arm.

Protein sequence alignment revealed that the evolutionarily conserved Y228 residue in GYG2 (Supplementary Fig. 4e) likely plays a role similar to that of Y195 in GYG1 as the glucose-receiving site[12]. Structural comparison also showed that Y195 in GYG1 interacts with neighboring Y197, which in turn interacts with D160 and W128 of a neighboring GYG1 molecule in the GS•GYG1 structure (Fig. 4e; PDB: 3T7O). These residues are conserved in GYG2, suggesting that both isoforms may share similar inter-subunit contacts to stabilize the catalytic sites (Supplementary Fig. 4f). Despite these conserved features, site-directed mutagenesis studies have showed that GYG2 has substantially reduced autoglycosylation activity[7,21,22]. Notably, the C loop in GYG2 is highly unstructured, whereas in GYG1, it contains a small α-helix (Fig. 4f, g). In GYG1, residue P238 within the C loop facilitates positioning of Y195 in the acceptor arm, enabling Y195 to attack the UDP-glucose moiety in the active sites. In contrast, this residue is replaced by S271 in GYG2 (Fig. 4g), thereby potentially disrupting this function. Therefore, the conformational differences in the C loop likely contribute to the reduced autoglycosylation activity observed in GYG2.

### Dynamic regulation of GYG1 and GYG2 interact with GS in response to metabolic changes

Our study observed altered glycogen content in the GYG1 knockout (KO), GYG2 KO, and double KO (DKO) lines (Fig. 1h). Moreover, a clinical study reported that, in two families with GYG2 deletions, 8 of 11 of the affected individuals had diabetes[17], a condition characterized by impaired blood glucose regulation. Although that study did not demonstrate a genetic link between GYG2 deletion and diabetes, the

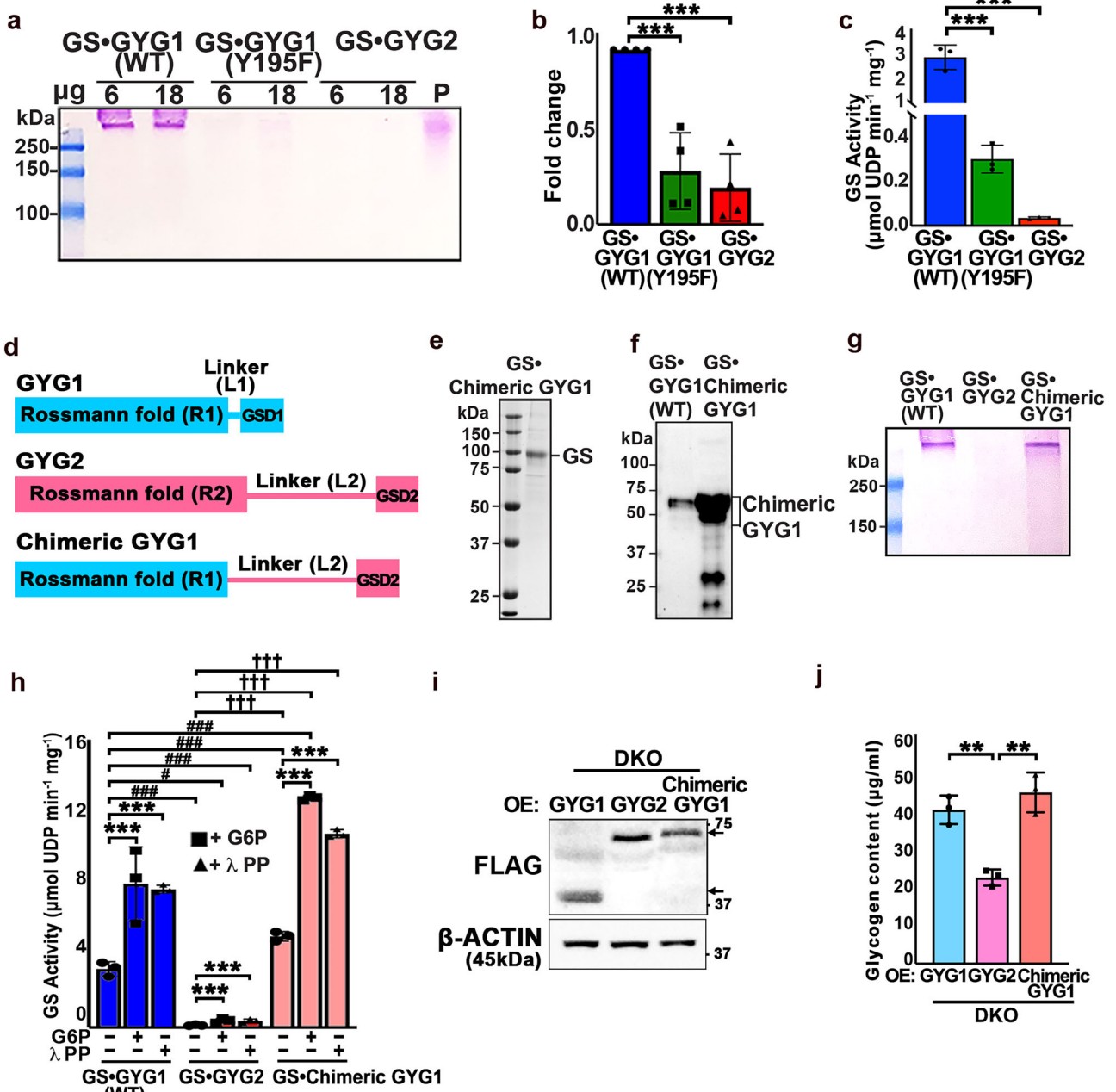

**Fig. 3 | GYG autoglycosylation is critical for GS activity. a** Representative PAS staining image of the GS•GYG1, GS•GYG1 (Y195F), and GS•GYG2 complexes. **b** Quantification of the band intensity in (**a**). Data are mean ± SD from four independent experiments. One-way ANOVA (Tukey's multiple comparison test); Statistical significance is indicated: *$P < 0.05$, ***$P < 0.001$. **c** GS activity of the GS•GYG1, GS•GYG1 (Y195F), and GS•GYG2 complexes. Data are mean ± SD from three independent experiments. One-way ANOVA (Tukey's multiple comparison test); Statistical significance is indicated: *$P < 0.05$, ***$P < 0.001$. **d** Domain structure of human GYG1, GYG2 and chimeric GYG1. **e** The GS•chimeric GYG1 complex purified by SEC (Superose 6) was analyzed by SDS-PAGE and stained with Coomassie blue. The data represent three independent experiments. **f** WT and GS•chimeric GYG1 complexes were analyzed by SDS–PAGE followed by Western blotting using anti-GYG1 antibody. The data represent three independent experiments.
**g** Representative PAS staining image of the GS•GYG1, GS•GYG2, and GS•chimeric GYG1 complexes. **h** GS activity of the GS•GYG1, GS•GYG2, and GS•chimeric GYG1

complexes under G6P or λPP treatment. Data are mean ± SD from three independent experiments. Two-way ANOVA (Tukey's multiple comparison test); Statistical significance is indicated: *$P < 0.05$, **$P < 0.01$, ***$P < 0.001$, # <0.05, ## <0.01, ### <0.001, †$P < 0.05$, ††$P < 0.01$, †††$P < 0.001$. Asterisks * represent statistical comparison relative to control of each complex plus G6P or λPP treatment. # represents statistical comparison of the GS•GYG2 and GS•chimeric GYG1 complexes with GS•GYG1 basal activity. † represents statistical comparison of GS•chimeric GYG1 with GS•GYG2 under G6P or λPP treatment. **i** Ectopic expression of FLAG-tagged GYG1, GYG2, and chimeric GYG1 in our DKO hESC lines. Protein levels of GYG1, GYG2, and chimeric GYG1-FLAG in DKO hESCs were analyzed by Western blot using anti-FLAG antibody. β-Actin served as the loading control. **j** Bar graph showing glycogen content in DKO hESC lines ectopically expressing GYG1, GYG2, or chimeric GYG1. Data are mean ± SD from three independent experiments. One-way ANOVA (Tukey's multiple comparison test); Statistical significance is indicated: **$P < 0.01$.

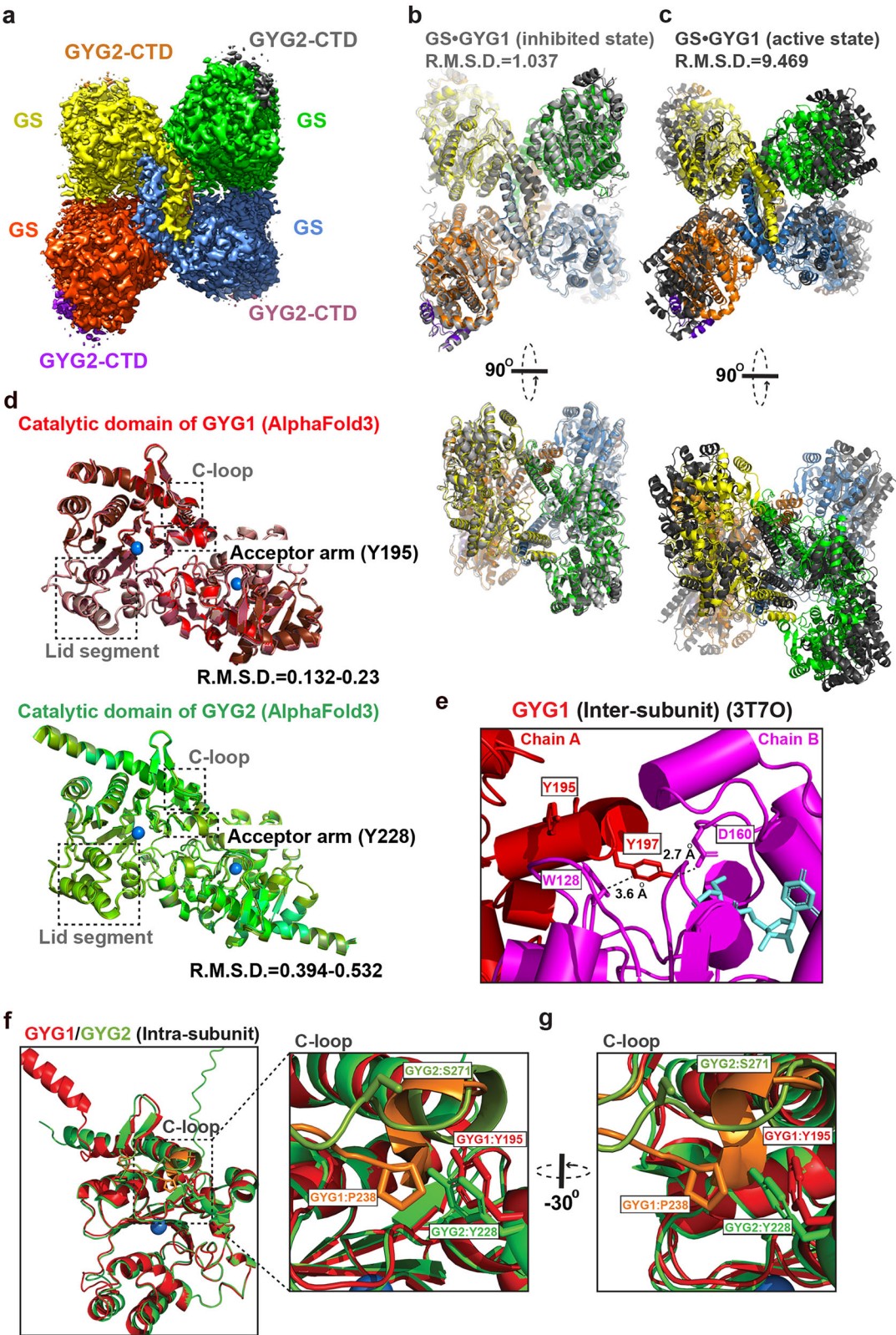

observed co-occurrence in several individuals raises the possibility of a metabolic association that merits further exploration. Notably, the study indicates that cellular glycogen content plays crucial roles in maintaining stable blood glucose levels and that GYG2 may be essential in responding to metabolic changes, affecting blood glucose concentrations. Building on these observations, we examined the expression levels of GYG1, GYG2, and GS in WT, GYG1 KO, and GYG2

KO cell lines under conditions of metabolic change, including during glycogen synthesis and breakdown. In WT cells, GYG1 was consistently highly expressed during both high glucose (which promotes glycogen synthesis) and forskolin (which stimulates glycogen breakdown) treatments (Fig. 5a, b). Notably, expression of GYG2 was initially low under the high glucose condition, but gradually increased over time (Fig. 5a, left panel). In contrast, during forskolin treatment, GYG2

**Fig. 4 | Structural characterization of the GS•GYG2 complex. a** Cryo-EM map of the heterotetrameric GS•GYG2 complex at 2.84 Å resolution. GS and the C-terminal domain of GYG2 (GYG2-CTD) subunits are colored separately. **b–c**, Structural superimposition of the GS•GYG2 complex with GS•GYG1 complex in the inhibited (**b**) or active (**c**) state. GS•GYG1 complexes are colored gray, and the color code of the GS•GYG2 complexes is as shown in (**a**). R.M.S.D. values are indicated. A 90°-rotated view is shown in the bottom panel. The PDB codes for GS•GYG1 complex in the inhibited and active states are 7Q0B and 7Q12, respectively. **d** AlphaFold3-predicted dimeric catalytic Rossmann fold domain of GYG1 (red) and GYG2 (green) across different species are superimposed, with R.M.S.D. values and manganese

(Mn) indicated. The lid segment, the acceptor arm, and the C loop are highlighted. **e** Cartoon representation of the crystal structure of the GYG1 catalytic domain. The autoglycosylation site (Y195) and Y197 that facilitate homodimeric interaction through contacts with D160 and W128 in an adjacent molecule are depicted. **f** Superimposition of AlphaFold3-predicted monomeric catalytic Rossmann fold domain of human GYG1 (red) and GYG2 (green) with the C loop region is emphasized. **g** A zoomed-in view of the C loop region of human GYG1 (red) and GYG2 (green) highlighted by the black dashed box in **f** is shown. Key residues Y195 and P238 in GYG1 and their corresponding residues in GYG2 (Y228 and S271) are shown.

expression was initially high, but then decreased (Fig. 5b, left panel). These results indicate that whereas GYG1 consistently supports glycogen synthesis, GYG2 acts as a metabolic sensor, adjusting glycogen regulation in response to energy demands. Furthermore, we observed that the expression pattern of GYG2 was closely correlated with phosphorylation of GS (pGS; inactive form) in WT cells (Fig. 5a, b). After 4 hours of high glucose treatment, total GS levels decreased, whereas pGS levels increased, indicating a feedback-driven slowdown in glycogen synthesis (Fig. 5a, left panel). This dynamic relationship between GYG2 and pGS was disrupted when GYG1 or GYG2 was knocked out (Fig. 5a, b), further supporting our findings.

To further confirm the interaction between GYG1 or GYG2 with GS or pGS, we conducted co-immunoprecipitation (co-IP) experiments to examine the interactions of GYG1 and GYG2 with GS under high glucose or forskolin treatments. To do so, we ectopically expressed GYG1 and/or GYG2 in the DKO line, denoted DKO^OE:GYG1 Flag (mimicking GYG2 KO), DKO^OE:GYG2 Flag (mimicking GYG1 KO), and DKO^OE:GYG1 Flag & GYG2 HA (mimicking WT), respectively. We found that GYG1 interacted with the active form of GS under high glucose conditions (Fig. 5c, upper panel). However, during the first hour of forskolin treatment, GYG1 interacted with pGS in DKO^OE:GYG1 Flag and DKO^OE:GYG1 Flag & GYG2 HA cells, indicating that GYG1-mediated glycogen synthesis activity is suppressed during glycogen breakdown. Notably, GYG2 only interacted with pGS under both the high glucose and forskolin treatments in our DKO^OE:GYG2 Flag cells. These findings suggest that GYG1 selectively binds active and inactive forms of GS, modulating glycogen synthesis activity in response to different metabolic conditions. In contrast, GYG2 remains in an inactive state, interacting solely with pGS during the late stage of glycogen synthesis and the initiating stage of glycogen breakdown. Thus, our results imply two layers of regulation in GYG-dependent glycogen synthesis: (1) by adjusting GYG expression levels; and (2) by modulating GYG binding to different phosphorylation states of GS.

### GYG1 and GYG2 deficiencies trigger distinct metabolic shifts that affect cellular glucose homeostasis

Glucose homeostasis is maintained by balancing glycogen storage and breakdown (e.g., through glycolysis). To assess the metabolic state of cells deficient in GYG1 or GYG2, we first measured their oxygen consumption rate (OCR) by means of Seahorse analysis. Our results show that the GYG1 KO lines exhibited significantly increased basal respiration, maximal respiration, and ATP production (Fig. 5d). In contrast, our GYG2 KO lines presented a reduced OCR, indicating a decrease in oxidative metabolism (Fig. 5d). Additionally, extracellular acidification rate (ECAR) analysis revealed that the GYG1 KO lines displayed lower glycolysis and glycolytic capacity (Fig. 5e), whereas the GYG2 KO lines relied primarily on glycolysis for energy production. These findings indicate that an absence of GYG1 reduces glycogen synthesis, forcing the cells to rely on alternative energy sources, whereas GYG2 deficiency causes glycogen accumulations, disrupting glucose metabolism and shifting energy reliance towards glycolysis. Together, these results support that both GYG1 and GYG2 regulate glucose homeostasis by influencing pathways involved in glycogen storage and breakdown.

Given the increased OCR and reduced ECAR observed in the GYG1 KO lines, we further investigated their metabolic phenotype using a metabolic flux assay. To do so, we treated the WT, GYG1 KO, and GYG2 KO lines with $^{13}C_6$-labeled glucose for 2 hr, before subjecting them to isotopomeric analysis (Fig. 5f, g). Heat map and isotopomer distribution analyses of multiple metabolic pathways revealed that the GYG1 KO lines exhibited reduced activity in the glycolytic pathway (i.e., levels of G6P, FBP, 2-PG, and pyruvate), but increased activity in the TCA cycle (i.e., levels of succinate, fumarate, and malate), relative to WT cells, whereas we observed the opposite scenario for the GYG2 KO lines (Fig. 5f, g). To further validate these findings, we treated WT, GYG1 KO, and GYG2 KO cells with 50 μM $^{13}C_{16}$-labeled palmitate for 16 hours. This treatment resulted in significantly increased levels of labeled TCA cycle metabolites (citrate/isocitrate, malate) in the GYG1 KO lines (Supplementary Fig. 5a, b). These results indicate that GYG1 KO cells preferentially rely on fatty acid oxidation for energy production rather than glycolysis. These results align with our findings of increased OCR and decreased ECAR in the GYG1 KO lines (Fig. 5d, e), further confirming that the metabolic shift in GYG1 KO lines is due to reduced glycogen synthesis and subsequent reliance on alternative energy sources. As GYG1 is involved in glycogen biosynthesis, its deficiency drives a shift toward oxidative metabolism to compensate for the lack of glycogen stores. In contrast, GYG2 KO lines favor glycolysis for energy production. Our findings indicate that GYG2 KO cells rely on glycolysis as their primary energy source, a metabolic adaptation similar to the shifts observed in diabetes[23], where impaired glycogen synthesis and glucose utilization lead to increased glycolytic activity. This similarity indicates that GYG2 KO cells may serve as a useful model for studying aspects of the metabolic dysregulation seen in diabetes.

### GYG2 controls glycogen particle size

Glycogen structure plays a critical role in regulating blood glucose levels. Beyond modulating glycogen content, GYG2 expression levels are also highly correlated with glycogen particle size across different tissues. For instance, GYG2 expression is absent or very low in skeletal muscle and neurons, where smaller glycogen particles predominate. In contrast, GYG2 expression is high in the liver, where larger glycogen particles are stored[10]. In humans, glycogen is organized into particles of different sizes, including α particles (up to 300 nm) and β particles (10-40 nm), with the former being assembled from the latter[24,25].

Based on these observations, we aimed to test if GYG2 plays a role in glycogen assembly. First, we isolated and purified glycogen from hESC lines of different genetic backgrounds, i.e., WT, GYG1 KO, GYG2 KO, and DKO. These cell lines exhibited varying levels of glycogen production, with the GYG2 KO and DKO lines producing notably more glycogen than the GYG1 KO line (Fig. 1h). Next, we conducted TEM on the glycogen particles in these cell lines, adjusting the quantity to ensure comparable particle densities on the TEM grids. In the WT cell line, we observed both compact and cauliflower-like α particles (54.8 ± 11.9 nm) and β particles (29.8 ± 5.4 nm) (Fig. 6a–c). Interestingly, glycogen particles in the GYG2 KO and DKO cells were more homogeneous and displayed sizes within the range of β particles (26.3 ±

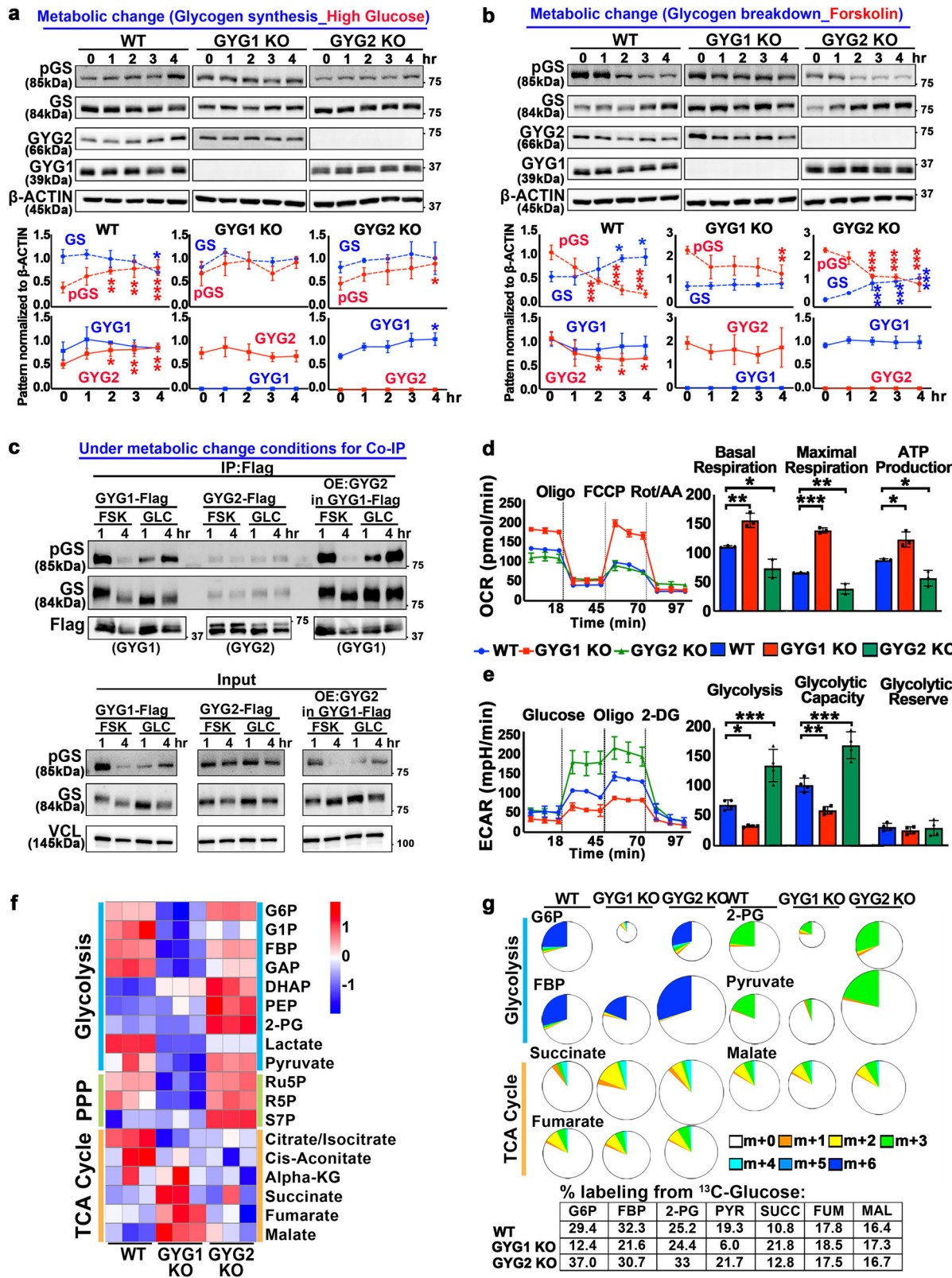

5.7 nm for the GYG2 KO and 26.0 ± 5.7 nm for the DKO lines) (Fig. 6a–c). In contrast, the GYG1 KO line exhibited generally smaller particles of two distinct populations: β particles (14.4 ± 3.2 nm) and smaller β particles (7.7 ± 1.6 nm) (Fig. 6a–c). Furthermore, the particle size distribution in the WT (α: 34.1 ± 2.3%; β: 65.9 ± 2.3%) differed markedly from that in the GYG1 KO line (β: 61.0 ± 6.8%; small β: 39.0 ± 6.8%) (Fig. 6c). Size exclusion chromatography linked to

multi-angle laser light scattering (SEC-MALLs) analysis confirmed that the glycogen particle sizes in the WT line were generally larger than those in the GYG2 KO, DKO, and GYG1 KO cell lines (Fig. 6d). Additionally, the GYG1 KO cell lines exhibited the smallest particle sizes and the fewest particles compared to the other three lines (Fig. 6d).

Furthermore, we ectopically expressed *GYG1* and *GYG2* in the DKO line, denoted DKO^OE:GYG1 and DKO^OE:GYG2, respectively. Upon

**Fig. 5 | Metabolic reprogramming in GYG1 and GYG2 KO cells reveals distinct energy pathways and dynamic regulation of their interactions with GS under metabolic change. a, b** Western blot analysis of WT, GYG1 KO, and GYG2 KO hESCs following metabolic changes. For (**a**), hESCs were starved and then treated with 50 mM glucose; for (**b**), cells were treated with 10 µM forskolin in starvation medium. Cell lysates were α-amylase–treated. β-Actin served as a loading control. Quantification is shown below each blot. Data are mean ± SD ($n = 3$) One-way ANOVA (Tukey's multiple comparison test); Statistical significance was compared to the 0-hr time-point: *$P < 0.05$, **$P < 0.01$, and ***$P < 0.001$. **c** Co-IP followed by Western blot analysis of pGS (inactive) and GS (active) from DKO$^{OE:GYG1-FLAG}$, DKO$^{OE:GYG2-FLAG}$, and DKO$^{OE:GYG1-FLAG\&GYG2-HA}$, under sequential forskolin and glucose treatment. Cell lysates were pretreated with α-amylase before FLAG pulldown. Vinculin served as the loading control. ($n = 3$). **d** Seahorse analysis of oxygen consumption rate (OCR) to assess basal respiration, maximal respiration, and ATP

production in WT, GYG1 KO, and GYG2 KO hESCs ($n = 3$). Data are mean ± SD. One-way ANOVA (Tukey's multiple comparison test); *$P < 0.05$, **$P < 0.01$, and ***$P < 0.001$. **e** Seahorse analysis of extracellular acidification rate (ECAR) to assess glycolytic capacity, glycolysis, and glycolytic reserve. Data are mean ± SD ($n = 4$). One-way ANOVA (Tukey's multiple comparison test); *$P < 0.05$, **$P < 0.01$, and ***$P < 0.001$. **f, g** Mass isotopomer distribution (MID) of [$^{13}C_6$] glucose-derived intermediates from glycolysis, the pentose phosphate pathway (PPP), and the tri-carboxylic acid (TCA) cycle in WT, GYG1 KO, and GYG2 KO hESCs. hESCs were cultured for 2 hr in medium containing 17.5 mM [$^{13}C_6$] glucose ($n = 3$). **f** Heatmap of the total labeling percentage of metabolic flux in WT, GYG1 KO, and GYG2 KO hESCs. Scale bar represents relative fold change of overall metabolites. **g** Pie chart size represents overall metabolite abundance, and the total percentage of labeled metabolites is displayed in the table at bottom.

overexpression of *GYG1*, the glycogen particles adopted a more compact and rounded morphology (Fig. 6e and Supplementary Fig. 6a). The cauliflower-like α particles were absent in the DKO$^{OE:GYG1}$ line, indicating that GYG2 may be required for α particle assembly (Fig. 6e; Supplementary Fig. 6a). Notably, when *GYG2* was overexpressed in the DKO$^{OE:GYG1}$ line (denoted DKO$^{OE:GYG1\&2}$), we observed the reappearance of cauliflower-like α particles (Fig. 6f; Supplementary Fig. 6b, 6c), implying that both GYG1 and GYG2 are involved in regulating glycogen particle morphology. Notably, excessive GYG2 led to smaller and the fewest particles overall (Fig. 6g; Supplementary Fig. 6d), likely due to a reduction in glycogen content. Thus, elevated GYG2 levels significantly reduce glycogen content, mimicking the effects observed for the GYG1 KO line.

To better understand the role of GYGs in the assembly of glycogen particles, we overexpressed chimeric *GYG1* in the DKO line (Supplementary Fig. 6e). This resulted in glycogen particles with a more compact and rounded morphology, similar to those observed in the DKO$^{OE:GYG1}$ cell line (Supplementary Fig. 6f). However, cauliflower-like α particles were also observed under high levels of chimeric GYG1 expression (Supplementary Fig. 6f). These results indicate that the Rossmann Fold Domain plays as a crucial role in regulating glycogenesis and that the GYG2 linker region may influence glycogen particle size. Moreover, overexpression of the *GYG1* Y195F mutant and the *GYG2* Y228F mutant in the DKO line (Supplementary Fig. 6g) led to a significant reduction in both glycogen particle size and glycogen content compared to the DKO line (Supplementary Fig. 6h, i).

### GYG1 and GYG2 orchestrate glycogen particle assembly in a cell-type-specific manner

Glycogen particle size varies across different tissues[10]. To further explore the roles of GYG1 and GYG2 in glycogen particle assembly across various cell types, we differentiated WT, GYG1 KO, and GYG2 KO hESC lines into hepatocytes, CMs, neurons, and skeletal muscle cells. This approach allowed us to investigate the cell-type-specific roles of GYG1 and GYG2 in glycogen metabolism. These hESC-derived cell lineages exhibited the expected morphologies of hepatocytes, CMs, neurons, and skeletal muscle, all of which expressed relevant lineage-specific markers: alpha-fetoprotein (AFP) for hepatocytes, TNNT2 for cardiomyocytes, beta-tubulin III (TUJ1) for neurons, and ACTN2 for skeletal muscle (Supplementary Fig. 7a). Furthermore, flow cytometry analysis revealed no significant differences in differentiation efficiency among these lineages (Supplementary Fig. 7a, right panel, and b). In the WT-derived cells, GYG2 was highly expressed in hepatocytes, and moderately in CMs, with relatively low expression in the neuronal and skeletal muscle cells (Fig. 7a). These expression levels are consistent with previously reported GYG2 levels in various cell types[17,21], validating that GYG2 expression is regulated in a cell-type-specific manner. Additionally, we observed cauliflower-like α particles in hepatocytes and CMs, which express both GYG1 and GYG2, but not in neurons or skeletal muscles, where GYG2 expression is absent or minimal

(Fig. 7b, c). These results also corroborate that GYG2 is involved in α particle assembly. Quantification of glycogen particle size also revealed that hepatocytes (α: 56.4 ± 15.2 nm; β: 29.6 ± 6.0 nm) hosted the largest particles, followed by CMs (α: 54.9 ± 13.2 nm; β: 28.5 ± 5.90 nm), and then neurons (β: 27.7 ± 5.7 nm) and skeletal muscles (β: 25.7 ± 6.1 nm) (Fig. 7b, c and Supplementary Fig. 7c).

Notably, unlike for GYG1 KO hESCs (Fig. 1h), the GYG1 KO-derived differentiated cell lineages still synthesized glycogen, albeit to a lower extent than WT lineages (Fig. 7d). Indeed, glycogen content was reduced in the hESC-derived cell lineages upon GYG1 knockout (Fig. 7d), whereas it was increased only in the GYG2 KO-derived CMs and hepatocytes, with no change observed for GYG2 KO neurons or skeletal muscle (Fig. 7d). Given that glycogen content was reduced in both hepatocytes and CMs of GYG1 KO cell lines, we further examined the expression levels of pGS and GYG2. Like GYG1 KO CMs, GYG1 KO hepatocytes displayed elevated levels of pGS and GYG2, leading to reduced glycogenesis (Fig. 7e, f). Notably, cauliflower-like α particles were observed in the WT-derived hepatocytes (WT-HEPs) (Fig. 7g), but they were absent from GYG1 KO HEPs and GYG2 KO HEPs (Fig. 7g). However, we detected α-amylase-resistant glycogen particles in GYG1 KO CMs, as shown by PAS staining (Fig. 7h). Moreover, TEM analysis revealed irregular particles with structures resembling polyglucosan bodies (Fig. 7i). Given that patients possessing GYG1 mutations display polyglucosan body accumulations in their CMs[26], our results indicate that a deficiency of GYG1 and concurrent overexpression of GYG2 can contribute to this pathological phenotype (Fig. 7i).

Neuronal cells produce less glycogen, so we had to use a larger quantity of neurons (5-8 times more cells compared to the other cell lines) to obtain comparable amounts of glycogen particles. Neuronal cells and skeletal muscle exhibited smaller particle sizes relative to the other cell types we assessed (Fig. 7b, c and Supplementary Fig. 7c). Despite minimal expression of GYG2 in skeletal muscle and neural tissues, there was no difference in glycogen particle size or morphology between the WT, GYG1 KO, and GYG2 KO lines (Fig. 7j, Supplementary Fig. 7d). Thus, GYG1 and GYG2 contribute to glycogen particle assembly in a cell-type-dependent manner. Our study indicates that human glycogenins regulate glycogenesis by modulating glycogen content, particle size, and glucose metabolic pathways, thereby maintaining cellular energy homeostasis. Furthermore, we postulate that disruption of this regulatory mechanism may contribute to the development of various metabolic disorders in humans.

We observed that levels of GYG2 were increased in the GYG1 KO CMs, potentially contributing to polyglucosan body formation (Fig. 7i). A previous study identified PPARG as the transcription factor regulating GYG2 expression[27]. Therefore, we sought to suppress GYG2 activity by treating GYG1 KO CMs with the PPARG antagonist GW9662 for 24 hours. Notably, treatment with 10 or 30 µM GW9662 significantly decreased levels of both pGS and GYG2 (Fig. 7k). Importantly, the GW9662 treatment prevented polyglucosan body accumulations in GYG1 KO CMs (Fig. 7l). Thus, the GYG2

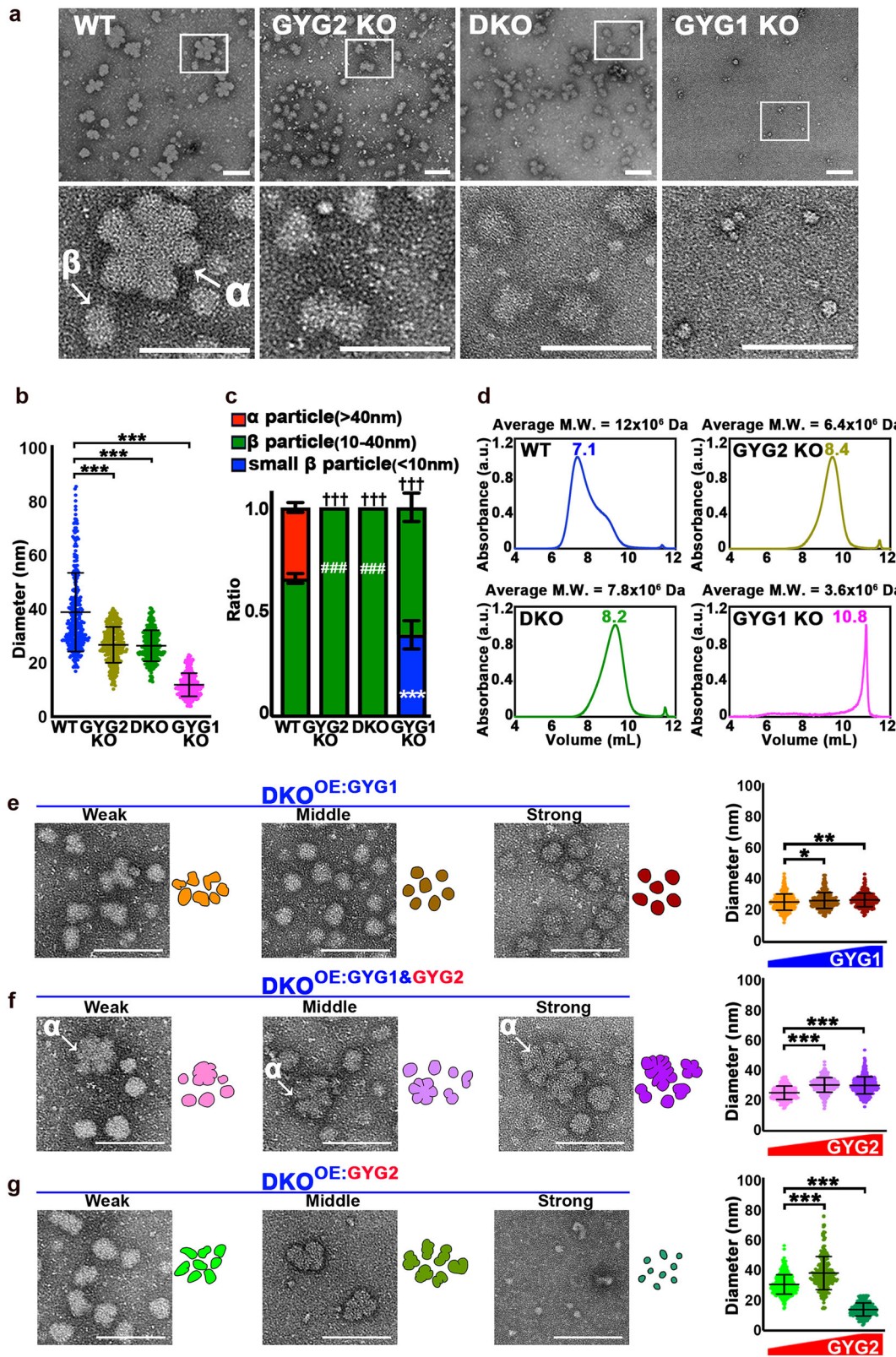

and pGS interaction in cells plays a critical role in modulating glycogen content.

## Discussion

Key findings of this work lie in clarifying the ongoing debate regarding the necessity of GYG in glycogen synthesis. Previously, GYG1 was thought to be potentially significant in glycogen metabolism, as its mutation has been linked to GSD XV. Conversely, GYG2 was thought to be functionally redundant, as glycogen synthesis was reported for *Gyg*-knockout mice (which possess a sole Gyg isoform)[18] and in human patients lacking GYG2[17]. Our study reveals the distinct roles and interplay between human GYG1 and GYG2 in glycogen metabolism. We reveal that GYG1 knockout in hESCs leads to an increase in GYG2, but reduces GS activity and glycogenesis. This finding suggests a nuanced

**Fig. 6 | Glycogen particle size is reduced in GYG2-free cell lines. a** Representative transmission electron microscopy (TEM) images of glycogen particles isolated from the indicated cell lines. Low and high magnification images are shown in the upper and lower panels, respectively. Scale bars: 100 nm. **b** Plot of glycogen particle diameter in the indicated cell lines. Data are mean ± SD from three independent experiments. One-way ANOVA (Tukey's multiple comparison test); Statistical significance: ***$P < 0.001$ ($n$ = at least 300 glycogen particles in total for each cell line). **c** Percentages of α ( > 40 nm), β (10-40 nm), and small β ( < 10 nm) particles isolated from the indicated cell lines, represented by red, green, and blue bars, respectively. Data are mean ± SD from three independent experiments. * represents a statistical comparison of each small β particle with WT. # represents a statistical comparison of each β particle with WT. † represents a statistical comparison of each α particles with WT. Differences were assessed statistically using two-way ANOVA (Tukey's multiple comparison test); ***$P < 0.001$, ###$P < 0.001$, †††$P < 0.001$ ($n$ = at least 300 glycogen particles in total for each cell line). **d** Glycogen particles isolated from the indicated cell lines were analyzed by SEC-MALLs. UV absorbance is measured in arbitrary units (a.u.). Elution volumes of the particles are indicated. The average molecular mass (M.W.) for the particles is shown above the plots. **e–g** Representative transmission electron microscopy (TEM) images of glycogen particles isolated from DKO lines subjected to three different concentrations (weak, middle, and strong) of GYG1 (**e**) or GYG2 (**g**). **f** Representative TEM images of glycogen particles isolated from DKO[OE:GYG1] hESCs subjected to three different concentrations (weak, middle, and strong) of GYG2. In **e–g**, cartoons illustrating the appearance of indicated glycogen particles are shown in the right side of indicated images; quantification analysis of particle sizes is presented in the panels at right ($n$ = at least total 300 glycogen particles in each cell line). Data are mean ± SD from three independent experiments. One-way ANOVA (Tukey's multiple comparison test); Statistical significance: *$P < 0.05$, **$P < 0.01$, and ***$P < 0.001$. All scale bars represent 100 nm.

and distinct role for GYG isoforms and GS activity, challenging previous assumptions about GYG2 redundancy in glycogen synthesis pathways. The suppressive effect of GYG2 on glycogenesis appears to arise from its limited autoglycosylation activity within the catalytic Rossmann fold domain. Furthermore, our data indicate that in the absence of GYG2, as is the case for our GYG2 KO and DKO hESC lines, glycogen synthesis homeostasis is disrupted, leading to higher levels of glycogen accumulation. Thus, GYG2 may play a role in maintaining balanced glycogen synthesis.

Our mechanistic analyses have revealed that the autoglycosylation activity of the GS•GYG2 complex is even lower than that of the GS•GYG1 (Y195F) complex, as determined by biochemical analyses. Furthermore, a GS•chimeric GYG1 complex presented re-activated autoglycosylation activity, indicating that the catalytic domain of GYG2 cannot function effectively as a glycogen primer. This deficiency in autoglycosylation activity likely contributes to the observed reductions in glycogen content displayed by GYG1 KO or GYG2-overexpressing hESCs. Notably, a previous study reported that GYG2 exhibits significantly limited glycosylation capacity, with -90% of GYG2 molecules remaining unglycosylated or being capable of binding only a single glucose unit[22]. Moreover, co-expression of GYG1 and GYG2 partly restored the glycosylation capacity of GYG2, with a maximum of only four detected glucose units compared to the 4-8 units that GYG1 could bind[22].

GS activity is classically regulated via phosphorylation by various kinases, including glycogen synthase kinase 3 (GSK3)[28], AMPK[29], protein kinase A[30], and casein kinase 2[20] or through allosteric activation by G6P[31]. Intriguingly, we observed a significant increase in phosphorylated GS (i.e., the inactive form) in GYG1 KO or GYG2-overexpressing lines, implying a potential regulatory role for GYG2 in modulating glycogen synthase activity. However, GYG2 lacks kinase activity and cannot directly phosphorylate GS, though we cannot exclude the possibility that GYG2 facilitates kinase-dependent GS phosphorylation. Moreover, our cryo-EM structural analysis has provided important insights into the interaction between GYG2 and GS. Consistent with previous findings[12], we observed that GS adopts an inactive conformation when bound to GYG2, hindering its activation. Similar structural behavior was observed for the GS•GYG1(Y195F) complex[12], which cannot autoglycosylate and likely assumes a more constrained conformation, thus also exhibiting reduced GS activity. Taken together, these findings support a model in which GYG2 modulates glycogenesis through two mechanisms: (1) its low autoglycosylation activity; and (2) by confining GS configuration to the inactive state.

Although the GYG2–GS interaction appeared relatively weak in our Co-IP experiments under metabolic changes (Fig. 5c), this is not unexpected for a regulatory modulator. All binding assays were conducted under matched conditions, and the relative interaction strength was consistently lower than that of GYG1. Nonetheless, GYG2 overexpression led to significant increases in pGS levels and reduced glycogen accumulation, indicating that even low-affinity or transient interactions can exert meaningful functional effects. This scenario is consistent with its proposed role in stabilizing GS in an inactive conformation rather than serving as a scaffolding component. Our findings further suggest that GYG2 functions as a negative regulator of glycogen synthase by binding to GS and promoting its phosphorylation, thereby reducing GS activity. In the absence of GYG2, this inhibitory interaction is lost, potentially leading to partial restoration of GS activity. However, despite this potential increase, the metabolic profile in GYG2 KO cells indicates a stronger shift toward glycolysis rather than enhanced glycogen synthesis. This suggests that glucose utilization in GYG2 KO cells may be redirected to meet immediate energy demands rather than being stored as glycogen.

Beyond its role in glycogen priming, GYG2 has recently been implicated in broader aspects of metabolic regulation. A transcriptome-wide analysis of human adipose tissue identified GYG2 as a mitochondria-associated gene under the transcriptional control of PPARG, with its expression positively correlating with mitochondrial function and thermogenic potential in subcutaneous adipose tissue[27]. Consistently, we found that pharmacological inhibition of PPARG reduced GYG2 expression and attenuated polyglucosan accumulation in GYG1-deficient cardiomyocytes. These findings indicate that a PPARG–GYG2 regulatory axis may modulate glycogen metabolism in a tissue-specific context. Although glycogen is not the primary energy reservoir in mature adipocytes, transient glycogen storage has been reported during early differentiation or insulin stimulation, implying a potential buffering role. The involvement of GYG2 in both glycogen structure and mitochondrial activity raises the possibility that GYG2 contributes to metabolic adaptability in energy-storing tissues, and its dysregulation may influence glucose handling in metabolic disorders. Moreover, the observed decline in GYG2 expression with age[27] implies a link between glycogen architecture and age-related metabolic dysfunction, although further studies will be needed to define this relationship in vivo.

In addition, a previous study indicated that non-glycosylated glycogenin serves as a binding agent linking β particles to α particles[4]. Remarkably, our study demonstrates that glycogen particles are significantly reduced in size in GYG2-free cell lines (i.e., our GYG2 KO, DKO, and DKO[OE:GYG1] lines), whereas they are smallest when GS•GYG2 was in excess (i.e., the GYG1 KO and DKO[OE:GYG2] lines). Consistently, a strong correlation between glycogen particle size and GYG2 expression has been reported across various human tissues/organs (e.g., brain, skeletal muscle, and liver)[5,10,22,26], with average particle size ranging from 10–40 nm in brain and skeletal muscle where GYG1 is the primary isoform, to 110–290 nm in the liver while GYG2 predominates[10]. However, we have observed that in the absence of GYG1 expression, the particles are very small and fewer in number. In contrast, in the absence of GYG2, we did not observe any cauliflower-like α particles, with the particles becoming rounded and less

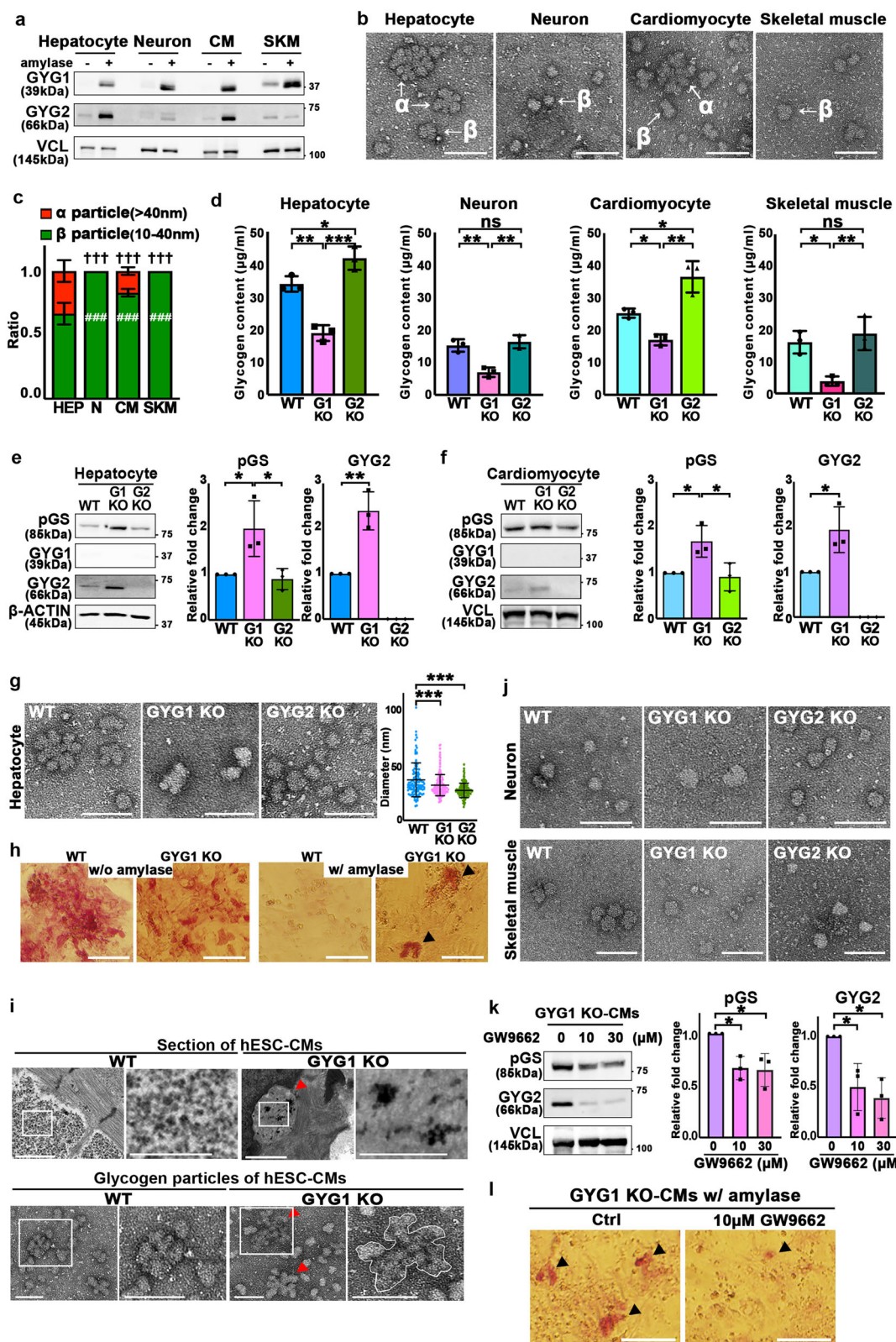

branched with increasing concentrations of GYG1. Hence, both GYG1 and GYG2 are essential for forming normal and appropriate amounts of glycogen particles. Thus, herein, we have demonstrated essential roles for both GYG1 and GYG2 in generating proper glycogen particles. Notably, some diabetic patients are known to possess glycogen particles of increased fragility and reduced stability, contributing to poor blood sugar control[32]. Furthermore, a previous study identified two

families in which GYG2 had been deleted, with a combined 8 out of 11 affected individuals being diabetic[17]. Although the study did not demonstrate clear genotype–phenotype segregation and clinical presentations varied, including cases requiring insulin treatment and others manageable through diet, the observed enrichment of GYG2 deletion among the diabetic individuals implies a possible association that warrants further investigation.

**Fig. 7 | An appropriate glycogen particle size distribution requires proper amounts of both GYG1 and GYG2 in different cell lineages. a** Western blots showing GYG1 and GYG2 levels from WT-derived hepatocytes, neurons, and skeletal muscles, with or without α-amylase treatment. Vinculin served as the loading control ($n = 3$). **b** TEM images of isolated glycogen particles. Scale bars: 100 nm. **c** Proportions of α (>40 nm) and β (10–40 nm) particles in the indicated cell types ($n = 3$). Data are mean ± SD ($n = 3$). †, and # represent statistical comparisons of α particles, and β particles in neurons, cardiomyocytes, and skeletal muscle relative to hepatocytes, respectively. Differences were assessed statistically using two-way ANOVA (Tukey's multiple comparison test); ### $P < 0.001$, ††† $P < 0.001$. **d** Glycogen content in indicated cell types. Data are mean ± SD ($n = 3$). One-way ANOVA (Tukey's multiple comparison test); *$P < 0.05$, **$P < 0.01$, ***$P < 0.001$. **e, f** Western blot analysis of pGS, GYG1, and GYG2 protein levels in cell extracts isolated from the indicated cell line-derived hepatocytes (**e**) and cardiomyocytes (**f**). β-Actin and Vinculin served as the loading control. Quantification analysis of pGS and GYG2 protein levels is presented in the panels at right. Data are mean ± SD ($n = 3$). Differences were assessed statistically using one-way ANOVA (Tukey's multiple comparison test). *$P < 0.05$, **<0.01. **g** TEM images and particle diameter quantification of glycogen in hepatocytes ($n = 100$–$150$ particles/cell line). Data are mean ± SD ($n = 3$). One-way ANOVA (Tukey's multiple comparison test); ***$P < 0.001$. Scale bars: 100 nm. **h** PAS staining of WT and GYG1 KO cardiomyocytes with or without α-amylase ($n = 3$). Scale bars: 200 μm. **i** TEM images of WT- and GYG1 KO-derived cardiomyocytes (upper) and glycogen particles (lower) ($n = 3$). Scale bars: 500 nm (upper) and 100 nm (lower). **j** TEM images of glycogen particles isolated from the indicated cell line-derived neuron cells (upper) and skeletal muscle (lower) ($n = 3$). Scale bars: 100 nm. **k** pGS and GYG2 levels in GYG1 KO cardiomyocytes treated with 0, 10, or 30 μM GW9662 (M6191, Sigma) for 24 h. Vinculin as control. Data are mean ± SD ($n = 3$). Differences were assessed statistically using one-way ANOVA (Tukey's multiple comparison test). *$P < 0.05$. **l** PAS staining of GYG1 KO-derived cardiomyocytes ± GW9662 treatment ($n = 3$). Scale bars: 100 μm.

Here, we found that GYG2 deficiency leads to the formation of smaller and less structured glycogen particles. Though glycogen synthesis can still occur in the absence of GYG2, these altered particles may be stored less efficiently and contribute to metabolic instability under conditions of increased glucose demand. This scenario is consistent with our mechanistic findings that GYG2 does not effectively prime glycogen synthesis and may act to constrain glycogen synthase activity through conformational modulation. Previous studies have reported that larger α-particles are associated with more stable glycogen storage, whereas smaller β-particles or fragmented forms are more susceptible to enzymatic degradation and contribute to unstable glucose homeostasis[10,20]. In our study, the reduction of α-particles and the formation of smaller glycogen particles in GYG1 KO, GYG2 KO, and DKO cells were accompanied by changes in total glycogen content (Fig. 1h), altered GS activity and phosphorylation (Figs. 2–3), and metabolic shifts toward glycolysis (Fig. 5d), further supporting a role for glycogen particle architecture in maintaining glucose homeostasis.

Taken together, these findings highlight a potential role for GYG2 not only in initiating glycogen formation, but also in regulating particle architecture and energy buffering (Fig. 8). Although the contribution of GYG2 deletion to diabetes susceptibility remains inconclusive, our data raise the possibility that loss of GYG2 may reduce metabolic flexibility and impair glucose homeostasis, particularly when additional genetic or environmental stressors are present. Future investigations using tissue-specific models or longitudinal metabolic profiling may help elucidate the broader physiological relevance of GYG2 in energy homeostasis.

## Methods

### hESC culture and directed cardiac cell differentiation
Human H9 ESCs were purchased from WiCell. All hESCs used in this study were derived from the MYH6:mCherry reporter line[33]. hESCs were grown on a Matrigel (Corning, 354277)-coated plate in Essential 8 medium (Gibco, A1516901) supplemented with E8 Supplement (Gibco, A1517101) and Normocin (InvivoGen, ant-nr-2) at 37 °C in 5% CO$_2$. The culture medium was exchanged daily. We used 0.5 mM EDTA (Invitrogen, 15575038) for routine passaging of hESCs.

### Directed cell lineage differentiation
We adopted a small molecule method for cardiac cell differentiation[34]. In brief, hESCs were cultured in hESC medium until they attained ~80–90% confluency. At differentiation day 0, hESCs were treated with 12 μM CHIR99021 (CHIR, Tocris Bioscience) in RPMI (Gibco, 11875085) supplemented with B27 minus insulin (Gibco, A1895601) and 100 U/ml Penicillin/Streptomycin (Corning, 30002CL) for 24 h. CHIR was removed the next day. At day 3, differentiated cells were treated with 5 μM WNT antagonist I (IWR-1, Tocris Bioscience) for 2 days. IWR-1 was removed at day 5. At day 7, B27 minus insulin in cardiac differentiation medium was changed to RPMI-B27 (Gibco, 17504044). Beating cells were typically observed around day 7. Cardiomyocytes were harvested from day 15 to day 50 for further analysis. For hepatocyte differentiation, hESCs were seeded at a dilution of 1:10–1:12. The following day, the cells were treated with 0.5% DMSO in E8 medium. The next day, cells were treated with 3 μM CHIR99021 in RPMI/B27 medium minus insulin for 24 h, followed by a switch to RPMI/B27 medium. From day 3 to day 8, cells were cultured in an Advanced F12 basal medium supplemented with small molecules A83-01, sodium butyrate, and dimethyl sulfoxide for 5 days. On day 9, differentiated cells were switched to Advanced F12 basal medium supplemented with small molecules, including FH1, FPH1, A83-01, dexamethasone, and hydrocortisone[35]. Hepatocytes were harvested from day 10 to day 15 for further analysis. Neuron differentiation was performed according to a neural differentiation method using PSC Neural Induction Medium (Cat no: A1647801) and the B-27™ Plus Neuronal Culture System (Cat no: A3653401, Thermo Fisher Scientific), following the manufacturer's instructions. hESCs were seeded at a dilution of 1:5. The next day, cells were treated with PSC Neural Induction Medium for 7 days, with medium changes every other day. On day 8, neural progenitor cells were switched to B-27™ Plus Neuronal Culture Medium. Neural cells were harvested between days 10 to day 15 for analyses. For skeletal muscle differentiation, we utilized a skeletal muscle differentiation kit (AMSBIO) according to the manufacturer's instructions. hESCs were seeded at a 1:20 dilution in skeletal muscle induction medium and cultured for 6-8 days, with medium changes every other day. Subsequently, myogenic precursors were passaged at a 1:6 dilution in skeletal myoblast medium for an additional 6-8 days. Myoblasts were then transferred to myotube medium. Myotubes were harvested around day 10 for analyses.

### Generation of GYG knockout hESC lines using the CRISPR/Cas9 technique
Two sets of sgRNA sequences for each gene were designed using the website http://chopchop.cbu.uib.no. The sgRNA sequences were ligated into pX330-U6-Chimeric_BB-CBh-hSpCas9 vector (Addgene, plasmid #42230)[36,37]. MYH6:mCherry hESCs were disassociated into single cells by incubating with Accutase for 5–7 mins. Cells were transfected by electroporation using an Amaxa Human Stem Cell Nucleofector Kit 2 (Lonza, VPH-5022) according to the manufacturer's guidelines. In brief, 1 million cells were resuspended in 100 μl nucleofector mix, which contained 4 μg CRISPR targeting plasmid. After electroporation, cells were replated on Matrigel-coated plates with 10 μM ROCK inhibitor (Tocris) at a low cell density (100 cells/well of a 6 well-plate). After ~10 days, single clones were picked and separated into two wells of a 96 well-plate. When the cells reached 80-90% confluency, we added lysis buffer (Sigma, Cat. #L3289) and neutralization buffer (Sigma, Cat. #N9784) to isolate genomic DNA for genotyping and DNA sequencing. All primers are listed in Supplementary Table 2.

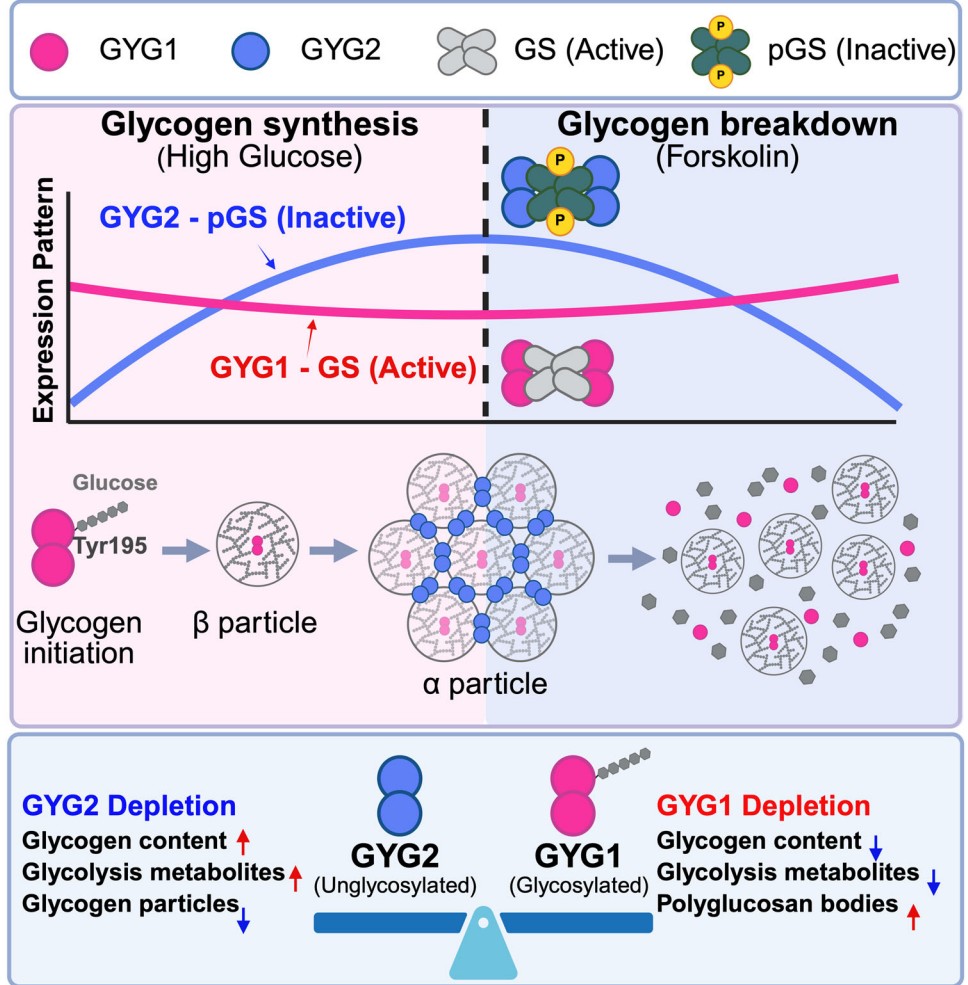

**Fig. 8 | Physiological relevance of dynamic regulation of GYG1 and GYG2 and their interactions with GS in glycogen metabolism.** GYG1 and GYG2 play distinct roles in glycogen metabolism. GYG1 functions as a glycogen primer through its autoglycosylation activity, promoting glycogen synthesis and the formation of glycogen β particles. GYG2, however, lacks autoglycosylation capability and cannot compensate for GYG1's role. Instead, GYG2 binds to glycogen synthase (GS) and enhances its phosphorylation, leading to decreased GS activity and reduced glycogen synthesis. Notably, in addition to regulating glycogen synthesis, GYG2 expression also controls glycogen particle size, as the formation of glycogen α particles requires both GYG1 and GYG2. Thus, the dynamic expression of GYG1 and GYG2 under metabolic change is crucial for maintaining glycogen balance. In the absence of GYG1 or GYG2, the regulatory mechanisms for glycogen balance are disrupted, leading to metabolic reprogramming and altered glycogen content, which affects normal physiological function. This model highlights GYG2 as a negative regulator of glycogenesis by modulating GS activity and influencing glycogen particle morphology. Created with BioRender.com.

## Ectopic expression in hESC lines

The human GYG1, GYG2, and chimeric GYG1 coding regions were amplified by polymerase chain reaction (PCR) using hESC cDNA as template and then cloned into the pCMV3flag8HOIL-1L vector (Addgene, #50016). The *GYG*-FLAG genes hosting FLAG tags were subcloned into lentivector pLKO AS3w.bsd. For virus production, 10 μg of target vector (GYG-FLAG), lentiviral envelope, and the packaging plasmids 12 μg psPAX2 (Addgene, #12260) and 6 μg pMD2.G (Addgene, #12259) were all transfected into 293 T cells by means of a standard CaCl₂ transfection method. Medium was changed the next day, and after 48 h, lentiviral supernatants were collected and filtered through a 0.45 μm syringe filter. GYG1 and GYG2 double-knockout (DKO) and wild-type (WT) hESCs were infected with GYG-FLAG lentivirus using a spin infection method. After infection, the medium was replaced with E8 medium. Two days later, Blasticidin (6.67 μg/ml, Gibco, R21001) was used to select GYG2-FLAG-positive clones.

## Periodic acid-schiff (PAS) stain

Cells were fixed with 4% paraformaldehyde for 8 mins at room temperature and washed three times with 1X PBS. A Periodic Acid-Schiff staining kit (Abcam, ab150680) was used for PAS staining. In brief, cells were first treated with periodic acid for 5 mins and then washed four times with reverse osmosis (RO) water. Next, the cells were stained with Schiff's solution for 15 mins, and rinsed with 50 °C RO water for 1 min. The samples were maintained in 1X PBS at 4 °C.

## Immunofluorescence analysis

Cells were fixed with 4% PFA for 8 mins at room temperature (RT). Blocking was performed in 5% horse serum (Gibco) and 0.3% Triton X-100 (93443-500 ML, Sigma-Aldrich) in PBS for 1 h at RT, followed by incubation of primary antibodies (Mouse anti-glycogen, a kind gift from Dr. Hitoshi Ashida of Kobe University; mouse anti-AFP, sc-8399, 1:200; mouse anti-TUJ1, GTX631836, 1:1000; rabbit anti-TNNT2, abcam ab45932, 1:800; mouse anti-ACTN2, Invitrogen 710947, 1:500) overnight at 4 °C or for 2 h at RT. Antibodies were detected using Alexa-488 and Alexa-647-conjugated donkey secondary antibodies against mouse or rabbit (1:1000, Invitrogen). Nuclei were counterstained with DAPI. Images were obtained using a fluorescence microscope (Olympus IX83).

## Western blotting

Cell pellets were lysed using RIPA buffer containing 1% PMSF. Protein concentration was measured according to the BCA method using a Pierce BCA Protein Assay Kit (23225, Thermo Fisher Scientific). Cell lysates were treated with 0.1 units/µl α-amylase (A1031, Sigma) at 37 °C for 30 mins to detect the total-form proteins. Samples were resolved on 10-13.3% sodium dodecyl sulfate polyacrylamide gels, which were then transferred to PVDF membrane at 120 V for 150 mins. Membranes were blocked using 5% non-fat dry milk in 1X TBST at RT for 40 mins and then incubated overnight with primary antibody at 4 °C. The following antibodies were used: mouse anti-GYG1 antibody (Santa Cruz, sc-271109, 1:1000); mouse anti-GYG2 antibody (Santa Cruz, sc-134346, 1:1000); rabbit anti-glycogen synthase (Cell Signaling #3893, 1:1000); rabbit phospho-glycogen synthase (Cell Signaling #3891, 1:1000); mouse anti-glycogen synthase 2 (Santa Cruz, sc-390391, 1:1000); rabbit anti-beta-Actin (Taiclone, tcba13636, 1:10000); mouse anti-alpha-Tubulin (GeneTex, GTX628802, 1:1000); mouse anti-Vinculin (V9131, Sigma-Aldrich); and rabbit anti-flag (GeneTex, GTX115043, 1:1000). After appropriate washing with 1X TBST, the membranes were incubated with HRP goat anti-mouse IgG (Jackson ImmunoResearch, 115-035-003) or HRP goat anti-rabbit IgG (GeneTex, GTX213110) at RT for 40 mins. Chemiluminescent assay was performed using ECL substrates, and signals were detected with a FUSION Solo S chemiluminescence imaging system (Vilber).

## Flow cytometry

hESC-derived hepatocytes, neural cells, cardiomyocytes, and skeletal muscle were dissociated into single cells and fixed by CytoFix (BD, 554714) for 20 min on ice, then washed with PBS three times. After separate incubation with anti-AFP, TUJ1, TNNT2, or ACTN2-FITC antibodies for 40 mins at 4 °C., then washed with PBS three times, the cells were analyzed on a BD FACSCanto II system.

## Glycogen content assay

Glycogen content was determined according to a glycogen colorimetric method (Colorimetric Assay Kit II, Cat no: K648-100, BioVision Inc., Milpitas, CA) and by following the manufacturer's instructions. One million cells were collected and homogenized in 200 µl ddH$_2$O on ice for 10 mins. They were then transferred to a heat block at 100 °C for 10 mins. The cells were then spun at 20000 x $g$ for 10 mins to remove cell debris. Supernatants of cell lysates and glycogen standards were hydrolyzed into glucose at room temperature for 30 mins. The glucose was then oxidized to form an intermediate that reduces a colorless probe to a colored product, also conducted at room temperature for another 30 mins. Samples without a glycogen hydrolysis step were used as controls to assess the contribution of background glucose to reduced final products. The absorbance of glycogen products was measured at 450 nm with a SpectraMax Plus384 microplate reader (Molecular Device, Sunnyvale, CA). Glycogen content was calculated by subtracting the glucose background from all samples.

## Glycogen activity assay

The activity of GS•GYG1, GS•GYG1 (Y195F), GS•GYG2 and GS•chimeric GYG1 complexes was measured using a UDP-Glo GT kit (Promega) according to the manufacturer's instructions. To measure activity, 25 µl/well of each reaction containing 100 nM protein complex, 1 mM UDP-glucose, 30 units of lambda phosphatase (New England Biolabs), and 2 mM Glc6P (Sigma) in assay buffer (25 mM HEPES pH 7.5, 200 mM NaCl, 0.5 mM TCEP) was dispensed into 96-well assay plates. Following a 60-min incubation at RT, 25 µl of UDP-Glo Plus detection reagent was added (final assay volume: 50 µl/well) and, after a further 60 mins of RT incubation, luminescence was detected using a SpectraMax M3 (Molecular Devices).

## Glycogen extraction

The approach for glycogen extraction was modified according to a previous study[22]. Cell pellets were resuspended in 200 µl (30% w/v) KOH and placed on a heat block at 100 °C for 25 mins. Pre-chilled ethanol (600 µl) was added to each cell extract and stored overnight at −30 °C, before centrifuging the samples the next day at 20000 x $g$ for 10 mins. The resulting pellet was washed four times with 66% ethanol and finally with 95% ethanol, before being dried for 10 mins on a heat block at 60 °C. Pellets were reconstituted in 100 µl of ddH$_2$O and stored at 4 °C until needed for experiments.

## Negative staining and transmission electron microscopy (TEM)

Carbon-coated copper grids were discharged first before use. A drop of glycogen solution was placed on the grid for 3 mins. Excess sample was drawn away using filter paper. Then the grid was stained with one drop of 2% aqueous uranyl acetate for 1 min. After being placed in a desiccator overnight, the grid was examined under TEM. Images were captured at 100 kV using a transmission electron microscope (Talos L120C 1.10.0, Thermo Scientific™) equipped with a Ceta CMOS camera and acquisition software (TEM Imaging & Analysis, TIA, version 4.18). Glycogen particle quantification was performed in ImageJ (version 1.53k).

## Transmission electron microscopy (TEM)

hESC-derived CMs were replated on Matrigel-coated Aclar embedding film for 5 days. Cells were fixed with 2.5% glutaraldehyde, 2% paraformaldehyde, and 0.1% tannic acid in 0.1 M cacodylate buffer (pH 7.2) for 30 mins. Cells were then washed three times with 0.2 M sucrose and 0.1% calcium chloride in 0.1 M cacodylate buffer (pH 7.2), and then post-fixed in 1% OsO$_4$ in 0.1 M cacodylate buffer (pH 7.2) for 30 mins. Cells were washed with ddH$_2$O three times and stained with 1% uranyl acetate for 30 mins. The cells were then washed for a further three times with ddH$_2$O before being dehydrated via an ethanol gradient (30%, 50%, 70%, 90%, 100%; for 5 mins at each concentration). Samples were infiltrated with 1:1, then 2:1 ratios of EPON:100% EtOH, and finally pure EPON resin. All above-described procedures were performed at RT. Samples were embedded in EPON at 60 °C for 48 hr. After slicing, images of cell ultrastructures were captured using a Tecnai G2 Spirit TWIN electron microscope (FEI) equipped with a Gatan CCD Camera (794.10.BP2 MultiScan) and acquisition software DigitalMicrograph (Gatan).

## Size exclusion chromatography-multiple angle laser light scattering (SEC-MALLs)

SEC-MALLs was performed on a WTC-100S5 7.8×300 mm column (Wyatt Technology). Molecular weight determination through MALLs involved utilizing a system comprising a multi-angle laser light-scattering photometer (DAWN HELEOS II; Wyatt Technology), a differential refractive index detector (Optilab T-Rex; Wyatt Technology), and a generic UV-absorbance detector that was prepared with 1X PBS containing 200 ppm sodium azide (Sigma-Aldrich 13412) at a flow rate of 0.5 mL/min. Samples (100 µL) were injected into a WTC-100S5 7.8×300 mm column, and data collection and processing were conducted using ASTRA software (Wyatt Technology). The differential refractive index increment (dn/dc) of glycogen was determined using the saccharide method (saccharide: 0.147).

## Insect cell expression constructs

The Multibac system was used to generate baculovirus strains expressing *Homo sapiens* GYG2. Full-length GYG2 was amplified by PCR and cloned into the pACEBac1 vector (ATG:biosynthetics GmbH). The Bac-to-Bac system was utilized to generate baculovirus strains expressing *Homo sapiens* GS1 (Uniprot ID P13807) (amino acids 1–737), GS2 (Uniport ID P54840) (amino acid 1-703), wild-type GYG1 (Uniprot ID P46976-2), the GYG1 Y195F mutant, wild-type GYG2 (Uniprot ID O15488-1), and the GYG2 Y229F mutant. Restriction sites necessary for cloning into the MultiBac and Bac-to-Bac vectors were included in the PCR primers. A

PreScission protease-cleavable hexahistidine tag was inserted in-frame at the N-terminus of GS, facilitating purification steps. The expression constructs were sequence-verified and transformed into DH10Multi-BacTurbo cells (ATG:biosynthetics GmbH) to generate bacmids.

## Protein expression and purification

Bacmids were transfected into Sf9 cells (Gibco) using FuGENE® HD Transfection Reagent (Promega). Baculoviruses were amplified twice, and fresh P3 virus from the GYG1 (WT), GYG1 (Y195F), and GYG2 bacmids were mixed with GS P3 virus for co-infection at a 1:1 ratio. Each virus was transfected into cells at a 1:10 ratio.

Cells were collected by centrifugation at $1000 \times g$ for 15 mins and resuspended in lysis buffer containing 50 mM K-phosphate (pH 7.4), 300 mM NaCl, 0.1% Tween-20, 3 mM β-mercaptoethanol, 5 % glycerol, 1 mM phenylmethylsulphonyl fluoride, and complete EDTA-free protease inhibitors (Roche). After sonication, the cell lysate was centrifuged for 30 mins at $35,000 \times g$. The supernatant was incubated with Ni-NTA resin (QIAGEN) for 30 mins. The Ni-NTA-fused protein was then eluted using lysis buffer containing 500 mM imidazole. Fractions containing the proteins were pooled and further purified using a 16/60 Superose 6 column. Protein quality was analyzed and confirmed by SDS-PAGE. Purified protein fractions were collected and concentrated to ~4 mg/ml using an Amicon® Ultra Centrifugal Filter, 30 kDa MWCO (Millipore), and stored in a buffer consisting of 20 mM HEPES pH 7.4, 150 mM NaCl, 3 mM DTT, and 5% glycerol at −80 °C. Glycosylated complexes were detected with PAS stain (Glycoprotein staining kit, Thermo Fisher Scientific).

## Native-mass spectrometry analysis of G6P-binding with protein complexes and decomposition of mass spectra

Protein samples were buffer-exchanged into 500 mM ammonium acetate buffer at pH 7.5, followed by mixing with G6P in a molar ratio of 1:0.025. The mixtures were immediately loaded into gold-coated emitters fabricated in-house and introduced into a modified Q-Exactive mass spectrometer (Thermo Fisher Scientific). A capillary voltage of 1.3 kV was applied during nanoelectrospraying, and an optimized acceleration voltage of 20 V was applied to the higher-energy collisional dissociation (HCD) cell following a gentle voltage gradient (injection flatapole, interflatapole lens, bent flatapole, and transfer multipole: 5.0, 4.0, 2.0, 1.0 V, respectively). Full mass spectra were acquired with a resolution of 25,000. RAW files were analyzed and manually processed using Thermo Xcalibur Qual Browser (version 4.4.16.14) in conjunction with UniDec (Universal Deconvolution) software (version 3.1.1.1). After UniDec processing[38], the m/z domain was deconvoluted into the mass domain to deduce G6P binding to the protein complex.

## Cryo-electron microscopy sample preparation

To prepare cryo-EM grids, the GS•GYG2 complex was diluted to ~2 mg/mL. Samples (4 µl) were loaded onto a glow-discharged holey carbon (Quantifoil R1.2/1.3) grid (Quantifoil Micro Tools GmbH). After blotting (3 sec with blot force 0), grids were plunge-vitrified into precooled liquid ethane using an FEI Vitrobot system (Thermo Fisher Scientific). The chamber was set to 4 °C and 100% humidity.

## Cryo-electron microscopy data collection

Cryo-EM grids were first checked on a 200 keV Talos Arctica transmission electron microscope equipped with a Falcon III detector (Thermo Fisher Scientific) operated in linear mode. The images were recorded at a nominal magnification of 120,000x, corresponding to a pixel size of 0.86 Å/pixel, with a defocus setting of −2.5 µm. Suitable cryo-EM grids were recovered for further data collection on a 300 keV Titan Krios transmission electron microscope (Thermo Fisher Scientific) hosting a K3 detector (with GIF Bio-Quantum Energy Filters, Gatan) operating in super-resolution mode, with gun lens = 4, spot size = 4, and C2 aperture = 50 µm using EPU-2.7.0 software (Thermo Fisher

Scientific). The raw movie stacks were recorded at a magnification of 105,000×, corresponding to a pixel size of 0.83 Å/pixel (super-resolution 0.415 Å/pixel). The defocus range was set to −1.2 ~ −1.8 µm and the slit width of energy filters was set to 15 eV. Fifty frames of non-gain-normalized tiff stacks were recorded with a dose rate of ~31 e-/Å2 per second and the total exposure time was set to 1.62 s, resulting in an accumulated dose of ~50.23 e-/Å2 ( ~1 e-/Å2 per frame). Images were taken with no tilt and 20-degree tilt of the grid. The parameters for cryo-EM data acquisition are summarized in Supplementary Table 1.

## Single-particle image processing and 3D reconstruction

The non-tilt and 20-degree tilt super-resolution image stacks were combined, motion-corrected, and dose-weighted using Motion-Cor263, with a 7 × 5 patch and two-fold binning (resulting in a pixel size of 0.83 Å). The motion-corrected micrographs were then imported into cryoSPARC64 for further single-particle reconstruction. The contrast transfer function (CTF) was determined from the motion-corrected images using the "Patch CTF estimation" function in cryoSPARC. Selected particles were extracted with a box size of 384 pixels for further 2D classification. Poor 2D class averages were removed after multiple rounds of 2D classification, and the remaining particles were used for ab initio map generation, followed by 3D heterogeneous refinement. Poor 3D classes were removed via several rounds of 3D heterogeneous refinement, and particles in good 3D classes were merged before conducting an additional 2D classification to remove any remaining bad particles. The good particles were further refined by 3D non-uniform refinement-imposed D2 symmetry. Map sharpening and resolution estimation were automatically performed via 3D non-uniform refinement and local refinement in cryoSPARC64. The overall resolution was estimated using the Fourier Shell Correlation (FSC) = 0.143 criterion, and the local resolution was also calculated in cryoSPARC64. The 3D density maps were visualized in ChimeraX65. The procedures for single-particle image processing and details of cryo-EM reconstruction are summarized in Supplementary Fig. 3. Statistical information for cryo-EM reconstructions is summarized in Supplementary Table 1.

## Atomic model building and refinement

The coordinate model was constructed based on the GS•GYG1 complex (PDB ID: 8CVX), and refined using the Phenix software suite[39] and COOT[40]. The statistics of model validations refer to the final output files from the real space refinement in PHENIX. Residues with missing densities were not modeled. All statistics pertaining to data collection and model refinement are presented in Supplementary Table 1.

## Seahorse assay

Seahorse assay was performed using a Seahorse XFe24 Extracellular Flux Analyzer (Agilent) to measure the oxygen consumption rate (OCR) and extracellular acidification rate (ECAR) of our cell lines. hESCs ($4 \times 10^4$ cells/well) were seeded in a Matrigel-coated XF24 microplate in E8 medium with ROCK inhibitor. The medium was changed every day. On the day of the assay, the medium was replaced with assay medium (Agilent). Cells were incubated in an incubator without $CO_2$ for at least 1 h before initiating the test. For the Mito Stress Test, the assay medium was Seahorse XF DMEM basal medium supplemented with glucose (17.5 mM, Sigma) and GlutaMAX (2 mM, Gibco). OCR was measured in the basal state and in response to a final concentration of 1 µM oligomycin, 0.25 µM FCCP, and 0.5 µM rotenone/antimycin A. For the Glycolysis Stress Test, the assay medium was Seahorse XF DMEM basal medium supplemented with GlutaMAX (2 mM, Gibco). ECAR was measured in the basal state and in response to a final concentration of 10 mM glucose, 0.1 µM oligomycin, and 50 mM 2-DG.

## Sample preparation for stable isotope-resolved metabolomics

WT, GYG1 KO, and GYG2 KO hESCs were treated with 17.5 mM $^{13}C_6$-labeled glucose (Sigma, 389374) for 2 hr in SILAC Advanced DMEM/F-12 Flex (Gibco, A2494301) supplemented with 200 μg/ml bFGF, 65 mg/ml Vitamin C, 543 mg/ml $NaHCO_3$, 12.5 mM NaCl, 1X GlutaMAX, and 1X Penicillin/Streptomycin. For tracing $^{13}C_{16}$-labeled palmitate, WT, GYG1 KO, and GYG2 KO hESCs were treated with 50 μM $^{13}C_{16}$-palmitate (Sigma, 605573) for 16 hr. To collect the metabolites, cells were extracted with 100 μL of pre-cooled extraction solvent (acetonitrile:methanol:water = 4:4:2, v/v/v). The sample was shaken for 5 mins at 1000 rcf, and then centrifuged at 15000 rcf for 5 mins at 4 °C. Ninety microliters of the supernatant was collected and mixed with 10 μL of 5 mM medronic acid. The sample was then filtered and subjected to ultra-high-performance liquid chromatography–electrospray ionization tandem mass spectrometry (UHPLC-ESI-MS).

## UHPLC-ESI-MS parameters for stable isotope-resolved metabolomics

For metabolite analysis, an Agilent 1290 II UHPLC system coupled with an Agilent 6545XT QTOF unit (Agilent Technologies) was used. A BEH amide column (2.1 × 100 mm, 1.7 μm) (Waters, Milford, MA) was used for separation. The injection volume of the sample was 10 μL for PPP pathway metabolites and 3 μL for glycolysis and TCA cycle metabolites. The analytical column was maintained at 40 °C. The mobile phase was composed of solvent A (15 mM ammonium acetate and 0.3% ammonium hydroxide (28-30%) in water) and solvent B (15 mM ammonium acetate and 0.3% of ammonium hydroxide (28–30%) in 90% acetonitrile), with a flow rate of 0.3 mL/min. The gradient profile was as follows: 0–8 mins, 90–50% solvent B; 8–10 mins, 50% solvent B; 10–11 mins, 50–90% solvent B; and 11–20 mins, maintained at 90% solvent B. A jet stream electrospray ionization source was used for sample ionization. Negative electrospray ionization modes were applied with the following parameters: dry gas flow rate, 8 L/min; dry gas temperature, 300 °C; nebulizer pressure, 40 psi; sheath gas temperature, 325 °C; sheath gas flow rate, 11 L/min; capillary voltage, 3500 V; nozzle voltage, 1000 V; and fragmentor voltage, 120 V. The mass scan range was 70–1100 m/z for positive mode and 70–1100 m/z for negative mode. The acquisition rate was 2 Hz for both modes.

## Co-immunoprecipitation

DKO hESCs overexpressing GYG1-FLAG, GYG2-FLAG, or GYG2-HA overexpressed in GYG1-FLAG were cultured in one six-well plate. After treatment with 10 μM forskolin or 50 mM glucose for 1 or 4 hr, the cells were harvested using Accutase. The cell pellets were washed with 1X PBS and resuspended in ice-cold lysis buffer (50 mM Tris-HCl pH 7.5, 1 mM EDTA, 150 mM NaCl, 1% Triton X-100, 10% glycerol, 1 mM PMSF) containing a protease inhibitor cocktail (Roche) and 0.05 units/μl α-amylase at 37 °C for 60 mins. After lysis, the cell lysates were centrifuged to remove cell debris, and the supernatant was collected. A portion of the supernatant was transferred to a new tube to serve as input for co-IP. For sample pre-cleaning, Mag-Beads (TOOLS, TOPG/A-2) were placed on a magnetic stand and equilibrated with Mag-Beads wash buffer (PBS with 0.02% Tween 20). After washing the Mag-Beads at least three times, the cell lysates were added with the beads and mouse IgG, and then incubated for 2 hr at 4 °C with end-over-end rotation. For sample binding, Anti-FLAG® M2 Magnetic Beads (Sigma, M8823) were used. The beads were first equilibrated with FLAG bead wash buffer (50 mM Tris-HCl pH 7.5, 1 mM EDTA, 150 mM NaCl, 0.5% NP-40, 10% glycerol, 1 mM dithiothreitol, 1 mM PMSF). The pre-cleaned supernatant was then incubated with the equilibrated FLAG beads for 3 hr at 4 °C with end-over-end rotation. The beads were collected and washed with FLAG bead wash buffer. To elute the immunoprecipitate, 3x FLAG peptide (30 μg/ml, Sigma-Aldrich) was added to the beads, and the mixture was incubated in an orbital shaker at 4 °C for 50 mins. Sodium dodecyl sulfate polyacrylamide gel electrophoresis (SDS-PAGE) was performed for detection and to separate the FLAG-associated immunoprecipitate.

## Statistics and reproducibility

Statistical analyses were conducted using Prism (version 9). Data are shown as mean ±S.D. We used a Shapiro-Wilk test to determine data normality. For normally distributed data, one-way ANOVA (> 2-group data) or two-way ANOVA (> 2-group data and two variables) followed by Tukey's post hoc test were used to calculate statistical significance. For data that failed the normality test, a Kruskal-Wallis test (> 2-group data) followed by Dunn's post hoc test was performed. The notation for p values is as follows (or as indicated in the figure legends): $*P < 0.05$, $**P < 0.01$, and $***P < 0.001$. At least three independent experiments were analyzed. All representative PAS staining and immunofluorescence images are repeated at least three independent experiments with similar results. No statistical method was used to predetermine sample size. No data were excluded from the analyses. The experiments were not randomized. The investigators were not blinded to allocation during experiments or outcome assessment.

## Reporting summary

Further information on research design is available in the Nature Portfolio Reporting Summary linked to this article.

## Data availability

The data supporting the findings from this study are available within the manuscript and its supplementary information. The source data underlying panels in Figs. 1–7 and Supplementary Fig. 1, 2, 5, 6, and 7 are provided in a source data file. Protein structural coordinates and maps have been deposited in the PDB and the Electron Microscopy Data Bank (EMD) with the accession codes PDB 8Z0A, EMD-39700. Native mass spectrometry for G6P binding have been deposited at Zenodo. The metabolomics data have been deposited to MetaboLights repository with the study identifier MTBLS12586. All other data are available from the corresponding author upon request. Source data are provided with this paper.

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

## Acknowledgements

The pFastBac-GS1, pFastBac-GN1 (WT), pFastBac-GN1 (Y195F) plasmids were kind gifts from Prof. Elton Zeqiraj of the University of Leeds. We are grateful for the technical support and advice provided by Dr. Jung-Lee Lin and his team at the GRC Mass Core Facility of the Genomics Research Center, Academia Sinica, Taipei, Taiwan. We appreciate the support of the Imaging Core at the Institute of Molecular Biology, Academia Sinica. We also extend our gratitude to Drs. Ching-Hua Kuo, Huai-Hsuan Chiu, and the staff of the Tenth Core Laboratory, Department of Medical Research, National Taiwan University, for their technical support. We also thank the Technology Commons, College of Life Science, National Taiwan University, for technical support. This work was supported by the National Science and Technology Council (NSTC- 111-2628-B-002-027-MY3 to S.-Y.T.; NSTC-112-2320-B-001-006-MY3 to K.-C.H.), National Taiwan University (112L895405 and 113L893105 to S.-Y.T.), and Academia Sinica (AS-IVA-112- L05 to K.-C.H. and AS-CDA-111-L03 to H-Y.Y.).

## Author contributions

T.W., Y.P., and C.C. - design, collection, and analysis of data; Y.C., W.H., Y.L., T.H., and S.L. - collection and analysis of data; P.C. - performance of cryo-EM experiment; Y.T. - recombinant protein purification; Y.-A. C. and H.Y. - analysis of native mass spectrometry. Y.-C. C. and C.L.- sequence and structural analyses. S.T. and K. H. - conception and design of experiments, data collection and analyses, manuscript writing, and financial support.

## Competing interests

The authors declare no competing interests.
