## [Transparent Peer Review file · Nature Communications]

Human glycogenins maintain glucose homeostasis by regulating glycogen metabolism

Corresponding Author: Professor Su-Yi Tsai

Version 0:

Reviewer comments:

Reviewer #1

(Remarks to the Author)

In this article, Weng and colleagues propose to study the distinct roles of glycogenin (GYG)1 and GYG2 in the control of glycogen synthesis and degradation. Their aim is to gain a better understanding of the differential contributions of these two proteins in different cell types, which is particularly relevant in the context of glycogen storage diseases. Most studies of the mechanisms have been carried out in mouse models in which a single GYG is expressed. Here, they propose to achieve their objectives by working on human embryonic stem cells (hESCs) differentiated into different lineages using structural biology, metabolic analysis and biochemistry. They show different expression profiles for GYG1 and GYG2 and different interaction dynamics with Glycogen synthase (GS). They demonstrate that GYG1 and GYG2 interact respectively with the active and inactive forms of GS during glycogen synthesis and degradation. They conclude that GYG2 negatively regulates glycogenesis, by regulating the size of glycogen particles, while GYG1 supports glycogen synthesis under conditions of energy abundance. Their work demonstrates the distinct roles of GYG1 and GYG2 in glycogen metabolism in different human cells. The article is well written and the experimental strategy well designed. The study is exhaustive, ranging from the description of GYG expression to molecular mechanisms and analysis of the structure of proteins and glycogen granules. It is complemented by metabolic challenge experiments that provide new mechanisms for regulating glycogen metabolism. The amount of work is impressive and provides a comprehensive overview of the molecular regulation of glycogen, which is of major interest in the field. However, some aspects of the article could be strengthened:

- The authors used a CRISPR/Cas9 approach to delete GYG1 and GYG2 in hESC and worked on two cell clones, although 3 are often necessary. They show that glycogen content is greatly reduced in GYG1 KO cells, while no change is observed in GYG2 KO cells. They show that GYG2 is increased in GYG1 KO cells and conversely, it would be important to know the expression of GYG1 in GYG2 KO cells. The authors go on to say: "To test whether GYG2 compensates for the function of GYG1 in glycogen synthesis, we generated double knockouts 'without GYGs'", which is not logical given that in the absence of GYG1, glycogen levels are very low, suggesting that there is no compensation for glycogen synthesis. It would be more accurate to say that they wanted to examine potential regulation of GYG expression in the two KOs GYG1 and GYG2. Evidence of GYG1 expression in GYG2 KO is missing, although it is essential. Furthermore, these results show that in this model, the absence of GYG2 induces an increase in glycogen content but the basal level of glycogen in WT cells is too high to measure this effect. The authors could find other culture conditions to reduce the basal level of glycogen in WT cells and compare it with GYG2 KO cells to identify a potential increase in glycogen in the latter.

- The explanation of the results for GS. GYG1 (Y195F) is not clear and should be reformulated with more detail. In addition, we can question the relevance of also carrying out mutations in GYG2 (Y228). It would be informative to know whether a chimeric GYG1 protein not with the entire Rossmann fold of GYG2 but only a change in the Y228 region would produce the same effects.

- The authors indicate that GYG1 deficiency induces a shift towards oxidative metabolism to compensate for the lack of glycogen reserves. This mechanism would need to be completed by an analysis of other energy substrates such as lipids in order to determine how this mechanism occurs.

- The authors differentiate hESC in the different lineages. They should study and indicate in the article whether the absence of GYG1 and GYG2 affects the different differentiation processes.

- In this article, there are two aspects: the role of GYGs in the regulation of GS and their role in the assembly of glycogen

particles. It would be interesting if the authors could make the link between the two parts. For example, the analysis of glycogen particles in their other cellular models, such as the chimeric GYG1, would be informative.

Minor points:

- Indicate GS- binding domain (GSD) in the figure legend to Fig. 1a
- Indicate in the text the addition of G6P in the experiments shown in Fig. 2m.T
- The number of samples/experiments need to be indicated in figure legend of Fig. 5
- The authors must provide more details about the PPARG experiment
- The discussion is relatively short and deserves to be developed further.

Reviewer #2

(Remarks to the Author)

Glycogen is a key energy store in animals, and regulation of glycogen synthesis and glycolysis are key aspects of metabolic processes. Enzymes that regulate glycogen formation and its metabolism have been extensively studied for over 50 years, although the mechanistic insights, especially at the structural level, are still emerging. This manuscript provides insights into how GYG1 and GYG2 isoforms in particular work to regulate glycogen synthesis.

Weng et al. have confirmed that both glycogenin (GYG) isoforms are required for normal glycogen regulation, and that the effects of each isoform expression lead to specific results in glycogen formation in vitro and in cells. They also study the activity of glycogen synthase (GS) complexes with GYG in vitro and in cells.

GS-GYG complexes containing primarily GYG2 isoform suppress GS activity, and that complexes containing primarily GYG1 isoform encourage GS activity. However, they also show that both GYG isoforms are required for the formation of both alpha-particles and beta-rosettes in cells. The authors also present CryoEM data of the GS-GYG2 complex, and demonstrate that it appears to form the inactive conformation of the GS1 tetramer seen previously (PMID: 35690592 and PMID: 35835870).

The work presented here covers a wide range of techniques, including cell-morphology data, CryoEM, and studies of in-cell interactions. I am excited about several of the observations presented in this paper, namely the role of both GYG isoforms in glycogen particle morphology, the regulation of GS activity by altering GYG isoform availability, and the role of GYG isoform availability in regulating whole-cell metabolic states. However, there are a few notions presented that go beyond what has been established and therefore require further scrutiny and experimentation before publication is warranted.

Major points:

One major flaw is that the authors have only used one GS isoform, which is likely to be GYS1 (GS1). (As a side note, the authors don't make this clear in the manuscript and they should clarify this point.) However, in cells, GYG2 will interact and most likely function in a complex with GYS2 since they are both abundant in liver. Check the protein atlas, but also multiple papers on this topic (e.g., PMID: 9346895, PMID: 22248338).

- From the data presented I could agree that the GYG2 isoform interacts more with pGS than the GYG1 isoform, but I think that concluding that the phosphorylation of GS is the manner in which GYG2 inhibits GS is unlikely, particularly as the authors were able to show an effect on GS activity by titrating in GYG1 in an in vitro assay with no kinases present. Can the authors further clarify this?

- I am also unconvinced by their assertion that the decreased autoglycosylation activity of GYG2 relative to GYG1 is due to the relative flexibility of the region around the autoglycosylation site. They have compared a crystal structure of GYG1 in the presence of other cofactors with an AlphaFold model of GYG2 and have concluded that the crystal structure is more stable, when in fact if you were to compare AlphaFold models of both GYG1 and GYG2 dimers, they are likely to find that the region that they describe is similarly structured in both. Therefore, the authors need to reevaluate these models (preferably using AlphaFold 3) and clearly highlight the differences (if any).

- I found the section describing the dynamic regulation of GS and GYG isoform interactions in response to metabolic changes very interesting, but I do have an issue with the way that the data shown in Figure 5c are interpreted. Throughout the paper the group has used western blots with basal GS or phosphorylated GS (pGS) antibodies to show the switch between the phosphorylated and un-phosphorylated state, however in Fig. 5c they have decided that the GS antibody is also picking up pGS siGYGal. Cross-reactivity of the antibodies is possible, but if this is the case it should be addressed in all cases where these antibodies have been used, particularly in the lower section of the same panel, where the different migration of samples in the gel across the different reaction conditions has not been labelled as cross-reactivity of the antibody.

I also found some of the methods used a bit unclear. The GYG2 isoform is represented in Fig. 1b as a 66 kDa species and in Fig. 1a as a 501 residue construct, but the calculated molecular weight for the 501 residue GYG1 isoform on UniProt id 55 kDa. Was this a different isoform, or just a typo? The Uniprot ID is listed in the methods for GS1 and GYG1, but not the GYG2 construct.

I also noticed the presence of lambda phosphatase in the UDP-Glo assay buffer. If GYG2 is affecting the activity of GS1 through regulation of its phosphorylation, why would you include the phosphatase in the assay conditions?

Fig. 2n – native mass spec data. I am finding this hard to understand what the stoichiometry is. The authors need to mention (and list in a table) what the expected mass and the measured mass is for each sample.

Minor points:

Y-axis values are missing in Fig 2o.

References 20 and 38 appear to be duplicated.

Nilsson, J. et al. LC-MS/MS characterization of combined glycogenin-1 and glycogenin-2 enzymatic activities reveals their self-glycosylation preferences. *Biochim Biophys Acta* 1844, 398-405 (2014).

Reviewer #3

(Remarks to the Author)

Reviewer #4

(Remarks to the Author)

Weng et al.

In this study Weng and colleagues use a human pluripotent stem cell model to study the role of glycogenins in the regulation of glycogen biosynthesis and cellular glucose metabolism. First the authors use knockout and overexpression models to examine the role of GYG1 and GYG2 in glycogen storage and the regulation of glycogen synthase. Then they undertake protein structure analysis of the complex of both gene products with glycogen synthase to elucidate differences in this interaction between GYG1 and GYG2. They conduct cellular metabolomics analysis in knockout and overexpressing cell lines to demonstrate how the two glycogenins function during challenges to glucose homeostasis, and their influence on oxidative phosphorylation and glycolysis. They use electron microscopy and biochemistry to analyze effects of the knockouts on glycogen storage granules in undifferentiated cells and then in four types of differentiated cell. The overall conclusion is that the two glycogenins function together in a complex fashion to control glycogen metabolism.

This analysis of the function of the glycogenins in a human cellular model is potentially significant, since the GYG2 gene is primate specific and mouse models are uninformative regarding its role. A major strength of this study is its very thorough and in-depth approach incorporating cellular models, biochemistry, metabolomics, structural biology, and transmission EM. The work is certainly a significant contribution to the detailed knowledge of the function of these two genes in metabolic regulation. I am less convinced that the study has much new to teach us about disease, or that it has "implications for treating glycogen storage diseases and metabolic disease" (from the Abstract). Here the main novelty lies in results with GYG2 since the function of GYG1 has been widely studied and is reasonably well understood. Throughout the study, strong GYG2 phenotypes are only obvious in the context of GYG1 knockouts or overexpression (with one exception, glycogen particle size in ESC), both somewhat artificial conditions. This is not surprising given the data from human genetics: reference 16 concerns a family with a deletion on the X encompassing GYG2 and is cited by the authors to suggest that the deletion has something to do with diabetes, also occurring at a relatively high frequency in this family. However, reference 16 makes clear that the diabetes is not segregating with the deletion, and that further study from the authors of Ref 16 and others has revealed that roughly 1/100 males in the general population bear similar deletions (which does not rule out some role for the glycogenin GYG2 but is consistent with the relatively weak phenotype of GYG2 KO through most of the authors' analyses).

I am also uncertain about the cellular modelling strategy to elucidate disease mechanisms. It is clear from prior work and the authors' Figure 7 that the actions of glycogenins are highly cell type specific. In this regard it might have made more sense to focus the model on differentiated cells like hepatocytes or cardiomyocytes rather than undifferentiated ESC.

It would be helpful if the authors produced a summary figure indicating very clearly what their results show about the role of GYG2 in normal physiology, how this might be perturbed in disease states and in what diseases specifically, and what the implications for therapeutic intervention might be. If they could make this case more strongly, and it is not obvious to me that they have done so thus far, the impact of the study would be increased considerably.

Specific comments (page numbers refer to PDF):

1. Page 3 introduction. Use consistent terminology (GYG does not refer to a specific human gene, gene names (human) should be capitalized in italics. Mouse *Gyg* gene lower case italics).
2. Page 5 What is phenotype of human GYG2 knockout described in Ref 16 (essentially there is none, see also comments above).
3. Page 7 justify use of undifferentiated hESC as primary model instead of hepatocytes, muscle neurons.
4. Page 8 minimal GYG2 KO effects in 1d (compared to WT) are difficult to reconcile with a doubling in glycogen content in 1h, as are double KO PAS and glycogen stains in 1g

5. Fig 2e-differences in pGS/GS ratio in DKO/OE GYG2 are not convincing. Why doesn't ratio change in G2 KO? How to reconcile double KO effects in 2e with effects on glycogen content in 1h? What is actual variability in control cells of GS and PGS measurement (normalized values only are shown).
6. Figure 2m GS/GYG2 activity seems negligible under any condition, results do not justify the conclusion that the complex is involved in any physiological regulatory mechanism
7. How much GYG2 protein is added in 2o-the decline in activity is not large and it would be important to know if these results had any physiological significance.
8. Figure 2m and Figure 3c is GYG2 inactivating the GS or is it just that it does nothing in the absence of GYG1
9. Page 16-the reference cited here (16) states clearly that the large deletion encompassing GYG2 did not segregate with diabetes in the family under study. The inference that GYG2 has some link to metabolic dysregulation in diabetes is not supported by this study.
10. Figure 5a, b-I am not convinced that changes over time in GYG1 or 2 are significant, neither does it look like the changes in GS/pGS, which are real, are much influenced by the GYG2 knockout. I think the authors have overinterpreted these data.
11. Figure 5c Interaction with GYG2 is minimal under any conditions. OE somewhat enhances changes in co-ip with GYG1.
12. Figure 5d GYG2 KO enhances glycolysis. This is one of the stronger effects of the KO. But what does this mean?
13. Page 20-again the extension of these findings to diabetes is a stretch.
14. Figure 6f-it is difficult to interpret data on the combined OE of GYG1 and GYG2 that seems to restore production of cauliflower like particles to the DKO line. The situation is artificial; data on the proportion of alpha particles should be provided.
15. Page 22-the characterization of the differentiated cells is insufficient.
16. Figure 7 d, e, and f-effects of GYG2 KO are unremarkable.
17. Figure 7d indicates that the effects of GYG1 KO are very cell type dependent, a fact that questions the use of undifferentiated ES cells as a model for most of the work. Note also that these differentiated cells are almost certainly of fetal phenotype.
18. Can the authors demonstrate any consequences of reduction of particle size in the mutant lines?

Version 1:

Reviewer comments:

Reviewer #1

(Remarks to the Author)

In this revised version of the manuscript, Weng and colleagues have addressed most of the comments suggested by reviewer 1. They have performed new experiments and modified the manuscript following the recommendations, which strengthens the article.

Reviewer #2

(Remarks to the Author)

We thank the authors for their rebuttal. They have addressed almost all of the issues that we identified, and we are overall supportive of publication. However there are still some minor points to be addressed.

The authors have incorporated the liver-specific GS2-GN2 complex, and they have clarified the GS isoform used throughout the experiments. They have justified the use of phosphatase in activity assays in response to the reviewers. They have also clarified the text in the paper to no longer infer that GN2 is causing GS phosphorylation, which was one of our concerns.

They have also addressed our point regarding cross-reactivity of the GS and pGS antibodies in a sufficient manner.

They have removed the comparison between stability of an AlphaFold model and a crystal structure, and have instead compared like-for-like AlphaFold models. I also appreciate the identification of specific residues mediating the stability effect that they describe.

Some minor points to address:

- Fig. 4f is very unclear, the residues described are hard to see, and the labels are obscuring sections of the models. I would suggest a more zoomed-in view with clearly visible and labelled residues, as in Fig. 4g, rather than the full monomer images displayed.

- Why are the organisms shown in Figure. 4e different between the GYG1 and GYG2 models? Both human GN isoforms are shown, but the other orthologs should either compare GN1 and GN2 from the same organisms, or the orthologs could be removed entirely and the comparison could be limited to the human enzymes only. The Drosophila GN1 model is particularly unhelpful, as it does not closely follow the same fold as the human enzyme, and it obscures the rest of the models.

- The inter-subunit interaction that they show in Figure. 4g are convincing, but these residues are also conserved in GN2, so they cannot be used as further evidence that the GN1 dimer is more stable than the GN2 dimer. The observation of the

interactions may stand, but the text should be clarified to reflect this.

- In Fig. 1 GN2 is labelled as 66 kDa, but its predicted molecular weight is 55 kDa. They address this in a response to reviewers by saying that it runs differently on a gel due to electrophoretic mobility effects, but this would not change the actual mass, only the predicted mass range that it runs at on a gel. Unless the sample has been subjected to MS to determine the accurate mass in solution, it should be labelled with its true molecular weight and the clarification can be indicated by a note (perhaps with with an Asterix?).

Overall, I believe that the main points of the paper are supported by the evidence provided, and that the work is of good quality. The demonstration of structural data, in vitro assays, and cell biology work is thorough and ultimately convincing.

The paper addresses a key question in the field (i.e. the role of the two glycogenin isoforms in glycogen particle morphology and regulation).

Reviewer #4

(Remarks to the Author)
Weng et al. R1

In their revision the authors address most points raised in my first review. They add some additional data to the manuscript and to the rebuttal to support their claims, and they have revised the text to clarify some key points and to add some caveats to their interpretation of the results. The graphic abstract also enhances the clarity of the presentation. Although the data implicating GYG2 in metabolic disease are far from convincing, I take the authors' point that the genetics of diabetes is in general quite complex and that variants in GYG2 could in certain genetic backgrounds and under certain environmental conditions contribute to the phenotype.

There are a few instances in the rebuttal where the authors offer clarification in response to the review but do not modify the text. I think the points below could be addressed in the text of the manuscript to provide additional clarity.

1. Page 8 Figure 1

"In our study, we relied on glycogen content assays to obtain quantitative data, which demonstrated a significant increase in glycogen levels in the GYG2 KO condition (Figure 1h). The discrepancy between the minimal changes observed from PAS staining (Figure 1d) and the notable increase in glycogen content may reflect the limitations of PAS staining in detecting subtle differences in glycogen accumulation, particularly in early-stage metabolic changes."

2. Figure 2m and Figure 3c

"In this context, GYG2 functions as a non-productive binding partner that neither activates GS nor supports primer formation and may restrict GS activity by stabilizing a conformation that is less permissive to activation."

3. Figure 5d

"Instead, our findings indicate that GYG2 functions as a negative regulator of glycogen synthase (GS) by binding to GS and promoting its phosphorylation, thereby reducing GS activity. In the absence of GYG2, this inhibitory interaction is lost, potentially leading to partial restoration of GS activity. However, despite this potential increase in GS activity, the metabolic profile in GYG2 KO cells indicates a stronger shift toward glycolysis rather than enhanced glycogen synthesis. This scenario suggests that glucose utilization in GYG2 KO cells may be redirected to meet immediate energy demands rather than being stored as glycogen."

We have now revised the Discussion to clarify this mechanism and to better explain the observed metabolic shift in GYG2 KO cells.

I am not sure where this is addressed in the revised discussion.

4. Glycogen particles

"Previous studies have reported that larger α -particles are associated with more stable glycogen storage, whereas smaller β -particles or fragmented forms are more susceptible to enzymatic degradation and thus contribute to unstable glucose homeostasis (PMID:22248338 and PMID:29483195). In our study, the reduction of α -particles and formation of smaller glycogen particles in GYG1 KO, GYG2 KO, and DKO cells was accompanied by changes in total glycogen content (Fig. 1h), altered GS activity and phosphorylation (Fig. 2–3), and shifts in glucose metabolism toward glycolysis (Fig. 5d)."

Reviewer #1 (Remarks to the Author)

In this article, Weng and colleagues propose to study the distinct roles of glycogenin (GYG)1 and GYG2 in the control of glycogen synthesis and degradation. Their aim is to gain a better understanding of the differential contributions of these two proteins in different cell types, which is particularly relevant in the context of glycogen storage diseases. Most studies of the mechanisms have been carried out in mouse models in which a single GYG is expressed. Here, they propose to achieve their objectives by working on human embryonic stem cells (hESCs) differentiated into different lineages using structural biology, metabolic analysis and biochemistry. They show different expression profiles for GYG1 and GYG2 and different interaction dynamics with Glycogen synthase (GS). They demonstrate that GYG1 and GYG2 interact respectively with the active and inactive forms of GS during glycogen synthesis and degradation. They conclude that GYG2 negatively regulates glycogenesis, by regulating the size of glycogen particles, while GYG1 supports glycogen synthesis under conditions of energy abundance. Their work demonstrates the distinct roles of GYG1 and GYG2 in glycogen metabolism in different human cells. The article is well written and the experimental strategy well designed. The study is exhaustive, ranging from the description of GYG expression to molecular mechanisms and analysis of the structure of proteins and glycogen granules. It is complemented by metabolic challenge experiments that provide new mechanisms for regulating glycogen metabolism. The amount of work is impressive and provides a comprehensive overview of the molecular regulation of glycogen, which is of major interest in the field. However, some aspects of the article could be strengthened:

We thank the reviewer for their thoughtful comments and for recognizing the comprehensive nature of our study, as well as their appreciation of the extensive work we have undertaken. We are grateful for their positive feedback and have carefully addressed the areas of concern to further strengthen our manuscript.

The authors used a CRISPR/Cas9 approach to delete GYG1 and GYG2 in hESC and worked on two cell clones, although 3 are often necessary.

We thank the reviewer for this comment. We have now generated a third cell line to reinforce our findings. We have confirmed depletion of GYG1 and GYG2 in this line by Western blot and DNA sequencing (Extended Data Fig. 1a, b, f).

We observed consistent phenotypes across all three knockout cell lines. Specifically: (1) PAS staining showed a significant decrease in glycogen levels in GYG1 KO #3 hESCs, whereas GYG2 KO #3 and DKO #3 presented comparable results to WT cells (Extended Data Fig. 1c); and (2) a glycogen content assay also demonstrated that the GYG1 KO #3 cells displayed reduced glycogen content, whereas both the GYG2 KO #3 and DKO #3 cells exhibited higher levels than WT cells (Extended Data Fig. 1g). These new results confirm that GYGs are

involved in regulating glycogen synthesis.

They show that glycogen content is greatly reduced in GYG1 KO cells, while no change is observed in GYG2 KO cells. They show that GYG2 is increased in GYG1 KO cells and conversely, it would be important to know the expression of GYG1 in GYG2 KO cells.

We thank the reviewer for this comment. We measured glycogen content and detected a significant increase in both GYG2 KO and DKO cells compared to WT cells (Figure 1h). Moreover, we now include new Western blot data and quantification results showing the GYG1 and GYG2 protein levels in the GYG1 KO and GYG2 KO lines (Fig. 1e and Extended Data Fig. 1d, e). These results show that GYG2 levels are elevated in GYG1 KO cells, whereas GYG1 levels are reduced in GYG2 KO cells. We have incorporated these findings into the main text of the revised manuscript (Page 8, lines 5-8).

The authors go on to say: "To test whether GYG2 compensates for the function of GYG1 in glycogen synthesis, we generated double knockouts 'without GYGs'", which is not logical given that in the absence of GYG1, glycogen levels are very low, suggesting that there is no compensation for glycogen synthesis. It would be more accurate to say that they wanted to examine potential regulation of GYG expression in the two KOs GYG1 and GYG2.

We agree with the reviewer's suggestion. We have now revised the sentence accordingly (Page 8, lines 8-10).

Evidence of GYG1 expression in GYG2 KO is missing, although it is essential.

Acknowledged. We have now included the relevant data in the revised manuscript. Western blot and quantification results have been added to Figure 1e and Extended Data Fig. 1d, e, and they are described in the main text (Page 8, lines 5–8). These results show that GYG2 expression is elevated in GYG1 KO cells, whereas GYG1 expression is decreased in GYG2 KO cells. Please note that the samples used for these experiments were not treated with α -amylase, and thus represent unbound (free) GYG proteins.

Furthermore, these results show that in this model, the absence of GYG2 induces an increase in glycogen content but the basal level of glycogen in WT cells is too high to measure this effect. The authors could find other culture conditions to reduce the basal level of glycogen in WT cells and compare it with GYG2 KO cells to identify a potential increase in glycogen in the latter.

We thank the reviewer for this comment. We would like to clarify that the basal level of glycogen in WT cells did not interfere with our ability to detect differences in glycogen content. In Figure 1h, we carefully used equal cell numbers for both the WT and GYG2 KO conditions, and the assay provided sufficient dynamic range to distinguish changes in glycogen levels. Similarly, in Figure 3c, we used equal amounts of purified GS complexes to assess enzymatic activity,

which further demonstrated that the GS•GYG1 complex (functionally similar to GYG2 KO) exhibited higher glycogen synthase activity than the GS•GYG2 complex (similar to GYG1 KO). Thus, both our cell-based assays and in vitro experiments support the conclusion that the absence of GYG2 promotes glycogen synthesis, and that the measurements were not affected by the basal glycogen levels in the WT cells.

The explanation of the results for GS. GYG1 (Y195F) is not clear and should be reformulated with more detail.

We appreciate the reviewer's feedback. Residue Y195 serves as the autoglycosylation site for GYG1, playing a crucial role in initiating glycogen synthesis by transferring UDP-glucose onto itself. When Y195 is mutated to Phenylalanine (F), the ability to autoglycosylate is lost, preventing glucose from attaching at this site (PMID: 35690592, Marr et al., 2022). We have now included this explanation in the revised manuscript (page 5, lines 1-4).

In addition, we can question the relevance of also carrying out mutations in GYG2 (Y228). It would be informative to know whether a chimeric GYG1 protein not with the entire Rossmann fold of GYG2 but only a change in the Y228 region would produce the same effects.

This is a very good point and we appreciate this suggestion. We have now generated the same mutation in GYG2 (Y228F). After purifying the GS•GYG2 Y288F complex, we confirmed the presence of GS and GYG2 proteins by means of Western blots (Extended Data Fig. 2i, j). Notably, similar to the observation for the GS•GYG2 complex, GYG2 presented a single band, representing minimal autoglycosylation. Moreover, we examined the GS activity of the GS•GYG2 Y288F complex and found that its activity was comparable to that of the GS•GYG2 complex (Extended Data Fig. 2i). Meanwhile, no signal for the GS•GYG2 Y288F complex was detected by PAS staining (Extended Data Fig. 2k). These results indicate that although Y228 represents a putative autoglycosylation site in GYG2, the Y228F mutation did not reduce this activity, as was the case for Y195F in GYG1. Thus, Y228 in GYG2 does not mediate the autoglycosylation activity necessary for glycogen synthesis.

In addition, when we ectopically overexpressed GYG1 Y195F and GYG2 Y228F, respectively in DKO hESCs, surprisingly, glycogen content significantly declined in both DKO^{OE:GYG1 Y195F} and DKO^{OE:GYG2 Y228F} cells (Extended Data Fig. 6i), and the glycogen appeared as small beta particles (as in GYG1 KO cells) (Extended Data Fig. 6h), indicating that the autoglycosylation ability of GYG1 is crucial to beta particle formation.

- The authors indicate that GYG1 deficiency induces a shift towards oxidative metabolism to compensate for the lack of glycogen reserves. This mechanism would need to be completed by an analysis of other energy substrates such as lipids in order to determine how this

mechanism occurs.

We appreciate the reviewer's insightful suggestion. To investigate the potential shift in energy substrate utilization, we treated WT, GYG1 KO, and GYG2 KO hESCs with $^{13}\text{C}_{16}$ -palmitate for 16 hr and analyzed the levels of incorporation of the labeled carbon into TCA cycle intermediates. We observed a significant increase in the percentage of labeled citrate/isocitrate and malate in GYG1 KO cells, whereas GYG2 KO cells showed levels comparable to WT cells (Extended Data Fig. 5a, b). Combined with our findings from the $^{13}\text{C}_{16}$ -glucose tracing experiments, these results indicate that GYG1 deficiency induces a metabolic shift from glycolysis toward increased reliance on fatty acid oxidation. These new findings have been incorporated into the revised manuscript (Page 21, lines 13–18).

- The authors differentiate hESC in the different lineages. They should study and indicate in the article whether the absence of GYG1 and GYG2 affects the different differentiation processes.

We thank the reviewer for this important comment. To address this point, we examined lineage-specific marker expression in differentiated cell populations by means of flow cytometry (Extended Data Fig. 7a, b). The histograms show comparable percentages of FITC-positive cells across WT and the GYG1- or GYG2-knockout lines. Quantification of differentiation efficiency is now presented in Extended Data Fig. 7b of the revised manuscript. These results indicate that the absence of GYG1 or GYG2 does not significantly impact the differentiation potential of hESCs into hepatocytes, cardiomyocytes, neurons, or skeletal muscle cells. We have now included these results and respective interpretation into the revised manuscript (Page 25, lines 14–16).

Minor points:

- Indicate GS- binding domain (GSD) in the figure legend to Fig. 1a

We apologize for the oversight. We have now added this detail into the figure legend to Fig. 1a.

- Indicate in the text the addition of G6P in the experiments shown in Fig. 2m.

We thank the reviewer for this comment. We now provide this information in the figure legend to Fig. 2m.

- The number of samples/experiments need to be indicated in figure legend of Fig. 5

We apologize for this oversight. This information has now been added to the figure legend of Fig. 5.

- The authors must provide more details about the PPARG experiment

We thank the reviewer for this comment. In a previous study (PMID: 38062989, Ham et al., 2024), it was reported that PPARG can act as an upstream regulator of GYG2. Since we hypothesized that GYG2 could be a negative regulator of glycogen synthesis, we wanted to apply a PPARG antagonist to reduce GYG2 expression. From our experiment, we observed that lower levels of PPARG upon antagonist treatment indeed reduced the levels of pGS and GYG2, as well as the extent of polyglucosan body accumulations (Fig. 7k, l).

- The discussion is relatively short and deserves to be developed further.

We appreciate the reviewer's suggestion. In the revised manuscript, we have now expanded the Discussion to include the broader biological relevance of PPARG-mediated regulation of GYG2. Specifically, we now highlight recent transcriptome-wide findings showing that GYG2 is a PPARG-regulated gene involved in mitochondrial metabolism and thermogenesis in adipose tissue (PMID:38062989). This finding complements our experimental observation that treatment with the PPARG antagonist GW9662 reduced GYG2 expression and phosphorylated GS (pGS) levels, thereby preventing polyglucosan body formation in GYG1 KO cardiomyocytes (revised Fig. 7k–l). Together, these findings indicate that the PPARG–GYG2 axis modulates glycogen metabolism in a tissue-specific and metabolically-responsive manner. In addition, we also now elaborate on GYG2's regulatory mechanisms, its low-affinity interaction with GS, and its role in metabolic flexibility and glycogen particle architecture.

This expanded section is now included in the revised Discussion (page 31).

Reviewer #2 (Remarks to the Author):

Glycogen is a key energy store in animals, and regulation of glycogen synthesis and glycolysis are key aspects of metabolic processes. Enzymes that regulate glycogen formation and its metabolism have been extensively studied for over 50 years, although the mechanistic insights, especially at the structural level, are still emerging. This manuscript provides insights into how GYG1 and GYG2 isoforms in particular work to regulate glycogen synthesis.

Weng et al. have confirmed that both glycogenin (GYG) isoforms are required for normal glycogen regulation, and that the effects of each isoform expression lead to specific results in glycogen formation in vitro and in cells. They also study the activity of glycogen synthase (GS) complexes with GYG in vitro and in cells.

GS-GYG complexes containing primarily GYG2 isoform suppress GS activity, and that complexes containing primarily GYG1 isoform encourage GS activity. However, they also show that both GYG isoforms are required for the formation of both alpha-particles and beta-rosettes in cells. The authors also present CryoEM data of the GS-GYG2 complex, and demonstrate

that it appears to form the inactive conformation of the GS1 tetramer seen previously (PMID: 35690592 and PMID: 35835870).

The work presented here covers a wide range of techniques, including cell-morphology data, CryoEM, and studies of in-cell interactions. I am excited about several of the observations presented in this paper, namely the role of both GYG isoforms in glycogen particle morphology, the regulation of GS activity by altering GYG isoform availability, and the role of GYG isoform availability in regulating whole-cell metabolic states. However, there are a few notions presented that go beyond what has been established and therefore require further scrutiny and experimentation before publication is warranted.

We thank the reviewer for their positive comments and for acknowledging the breadth of techniques employed in our study, including cell morphology analysis, cryo-EM, and in-cell interaction studies. We are pleased that the reviewer found our observations regarding the roles of GYG isoforms in glycogen particle morphology, GS activity regulation, and whole-cell metabolic states particularly exciting. We greatly appreciate their encouraging feedback.

Major points:

One major flaw is that the authors have only used one GS isoform, which is likely to be GYS1 (GS1). (As a side note, the authors don't make this clear in the manuscript and they should clarify this point.) However, in cells, GYG2 will interact and most likely function in a complex with GYS2 since they are both abundant in liver. Check the protein atlas, but also multiple papers on this topic (e.g., PMID: 9346895, PMID: 22248338).

We thank the reviewer for this important observation. We confirm that the original data in our study utilized GYS1 (GS1) to investigate the regulatory interactions between glycogen synthase and glycogenins. As noted, GYS2 (GS2) is predominantly expressed in the liver and has been reported to interact with GYG2 (PMID: 22248338). To address this point, we purified the GS2•GYG2 complex (Extended Data Fig. 2e, f) and performed PAS staining, which showed no detectable polysaccharide signal (Extended Data Fig. 2g), similar to what we observed for the GS1•GYG2 complex (Fig. 3a, g), corroborating that GYG2 exhibits low autoglycosylation activity regardless of its binding partner. Furthermore, GS activity assays revealed that the GS2•GYG2 complex displayed comparably basal activity to that of the GS1•GYG2 complex (Extended Data Fig. 2h and Fig. 3h), reinforcing that GYG2 does not effectively initiate glycogen synthesis in complex with either GS isoform (Extended Data Fig. 2e–h) (Page 13, lines 2-12). We have now clarified our use of GS1 in the revised manuscript (Page 10, lines 1-3) and include these new findings to address the reviewer's concern.

- From the data presented I could agree that the GYG2 isoform interacts more with pGS than the GYG1 isoform, but I think that concluding that the phosphorylation of GS is the manner in

which GYG2 inhibits GS is unlikely, particularly as the authors were able to show an effect on GS activity by titrating in GYG1 in an in vitro assay with no kinases present. Can the authors further clarify this?

We thank the reviewer for this insightful comment. We agree that phosphorylation alone may not fully account for the suppression of GS activity by GYG2. In our GS activity assays (Fig. 2m), although λ PP or G6P treatment enhanced the activity of the GS•GYG2 complex by ~4-fold, its absolute activity remained markedly lower than that of the GS•GYG1 (i.e., WT) complex. This outcome suggests that additional mechanisms beyond phosphorylation are likely involved.

We propose that the interaction between GYG2 and GS induces a conformational state that is less permissive to activation. This interpretation is consistent with a recent cryo-EM analysis (PMID: 35690592), which showed that GS adopts a tense, inactive conformation when phosphorylated, and transitions to a relaxed, active state upon dephosphorylation or G6P binding. Notably, we observed similar behavior in the GS•GYG1(Y195F) mutant complex, which cannot autoglycosylate and assumes a more rigid structure, in that it also exhibits reduced GS activity even after λ PP treatment.

These observations support a model in which GYG2 may act by stabilizing a conformation of GS that disfavors activation, independently of phosphorylation status. Combined with the minimal autoglycosylation activity of GYG2, the GS•GYG2 complex exhibits substantially reduced glycogen synthesis activity.

To better reflect this interpretation, we have revised the sentence in the main text (Page 10, line 19 and Page 11, lines 1–3) as follows:

Original:

“These results corroborate that phosphorylation of the GS•GYG2 complex is involved in the regulatory mechanisms responsible for inhibiting GS activity.”

Revised:

“These results indicate that phosphorylation may contribute to the suppression of GS activity in the GS•GYG2 complex. However, the persistently lower enzymatic activity, even after λ PP or G6P treatment, indicates that GYG2 may regulate GS activity through additional mechanisms.”

This interpretation has also been incorporated into the revised Discussion (Page 30, lines 5–7).

- I am also unconvinced by their assertion that the decreased autoglycosylation activity of GYG2 relative to GYG1 is due to the relative flexibility of the region around the

autoglycosylation site. They have compared a crystal structure of GYG1 in the presence of other cofactors with an AlphaFold model of GYG2 and have concluded that the crystal structure is more stable, when in fact if you were to compare AlphaFold models of both GYG1 and GYG2 dimers, they are likely to find that the region that they describe is similarly structured in both. Therefore, the authors need to reevaluate these models (preferably using AlphaFold 3) and clearly highlight the differences (if any).

We thank reviewer for these suggestions. We have now reevaluated the models using AlphaFold3 and observed that the predicted catalytic Rossmann fold domain structures of GYG1 and GYG2 exhibited minimal RMSD values (0.132 to 0.23Å for GYG1 and 0.394 to 0.532 for GYG2), indicating conservation among the predicted structures (Fig. 4d). A previous study (PMID: 22160680) proposed that three key regions of GYG1—the lid segment, the acceptor arm containing the Y195 acceptor residue, and the C loop adjacent to the acceptor arm—undergo conformational rearrangements that modulate its activity.

Interestingly, we found that the C loop in GYG2 is highly unstructured, yet it contains a small α -helix in GYG1 (Fig. 4e). These structural features are evolutionarily conserved across species for GYG1 and GYG2, respectively (Fig. 4e). Notably, in GYG1, residue P238 within the C loop facilitates proper positioning of F195 in the acceptor arm, thereby enabling Y195 to attack the UDPG glucose moiety in the active sites. In contrast, this residue is replaced by S271 in GYG2 (Fig. 4f), potentially impairing its function.

In addition to the intra-subunit interactions that stabilize Y195, we observed that Y195 in GYG1 interacts with neighboring Y197, which further interacts with D160 and W128 of a neighboring GYG1 molecule in the GS•GYG1 structure (Fig. 4g; PDB: 3T7O). These intra-and inter-subunit structural features contribute to an optimal configuration for glucose reception by Y195, enhancing overall enzyme stability. Hence, GYG1 displays higher autoglycosylation activity than GYG2. We have incorporated this new model into our revised manuscript (Page 16, lines 13-19 and Page 17, lines 1-18).

- I found the section describing the dynamic regulation of GS and GYG isoform interactions in response to metabolic changes very interesting, but I do have an issue with the way that the data shown in Figure. 5c are interpreted. Throughout the paper the group has used western blots with basal GS or phosphorylated GS (pGS) antibodies to show the switch between the phosphorylated and un-phosphorylated state, however in Fig. 5c they have decided that the GS antibody is also picking up pGS signal. Cross-reactivity of the antibodies is possible, but if this is the case it should be addressed in all cases where these antibodies have been used, particularly in the lower section of the same panel, where the different migration of samples in

the gel across the different reaction conditions has not been labelled as cross-reactivity of the antibody.

We appreciate the reviewer's thoughtful comment regarding the interpretation of the data in Figure 5c. Throughout this study, we used the anti-glycogen synthase antibody (Cell Signaling #3893) and anti-phospho-glycogen synthase antibody (Cell Signaling #3891) to distinguish between phosphorylated and non-phosphorylated forms of GS. To minimize the possibility of cross-reactivity, Western blots for these two antibodies were performed consistently on separate membranes.

To directly address the reviewer's concern regarding Figure 5c, we have now additionally performed immunoblotting using the anti-pGS antibody (Cell Signaling #3891) on the same sample set. The new results confirm that the slower migrating bands detected by the anti-GS antibody in the samples correspond to the phosphorylated GS species. This outcome validates our original interpretation and confirms that the observed band shift reflects changes in GS phosphorylation rather than antibody cross-reactivity.

Accordingly, we have updated the figure legend of Figure 5c to clarify that both GS and pGS antibodies were used to confirm the phosphorylation state of GS in this experiment.

I also found some of the methods used a bit unclear. The GYG2 isoform is represented in Fig. 1b as a 66 kDa species and in Fig. 1a as a 501 residue construct, but the calculated molecular weight for the 501 residue GYG1 isoform on UniProt id 55 kDa. Was this a different isoform, or just a typo? The Uniprot ID is listed in the methods for GS1 and GYG1, but not the GYG2 construct.

We thank the reviewer for these comments and would like to clarify that both Fig. 1a and Fig. 1b represent the GYG2 construct, not GYG1. We used the human GYG2 isoform corresponding to UniProt ID O15488-1, which encodes a 501-residue protein with a theoretical molecular weight of 55 kDa.

In our immunoblotting using mammalian cell lysates and purified recombinant protein from insect cells, GYG2 consistently appeared as a ~66 kDa species, which is in agreement with the observed band size reported in the datasheet of the anti-GYG2 antibody (Santa Cruz, sc-134346). This upward shift is likely due to intrinsic properties of GYG2 or possible post-translational modifications that influence its electrophoretic mobility, as is commonly observed in SDS-PAGE analyses.

We have now corrected the Methods section to include the UniProt ID for GYG2 (O15488-1).

I also noticed the presence of lambda phosphatase in the UDP-Glo assay buffer. If GYG2 is affecting the activity of GS1 through regulation of its phosphorylation, why would you include the phosphatase in the assay conditions?

We appreciate this insightful observation. Since regulation of GS activity can encompass dephosphorylation by phosphatase or allosteric binding with G6P, we utilized lambda phosphatase to address the dephosphorylation state. Therefore, we wanted to ensure that these GS•GYG complexes could be activated by dephosphorylation. However, under this condition, the GS•GYG2 complex still exhibited the lowest GS activity.

Fig. 2n – native mass spec data. I am finding this hard to understand what the stoichiometry is. The authors need to mention (and list in a table) what the expected mass and the measured mass is for each sample.

In order to resolve small-molecule binding (~260 Da) to large GS•GYG complexes (>2000-fold difference in mass), we maximized the resolution of the mass spectrometer to 25,000. Under high-resolution measurement, the multiple signals resulted in partial overlap. In order to deconvolute these signals, we introduced the UniDec software for mass spectrum deconvolution by transforming the m/z dimension into the mass domain. Following peak-shape fitting using Gaussian models, we were able to determine the centroid values of individual peaks as defined average masses. The mass differences calculated from the deconvoluted data closely match the molecular weight of G6P (260.2 Da), supporting our interpretation of multiple G6P binding. The deconvoluted masses derived from the m/z spectrum are now summarized in Extended Data Table 4.

Please note that, while GYG2 and GYG1 (Y195F) display different molecular weights in SDS-PAGE (Fig. 2i, j), our native mass spectrometry analyses revealed observed molecular masses of 534,837 Da for the GS1•GYG2 complex and 534,845 Da for the GS1•GYG1 (Y195F) complex (Table 1 to Reviewer #3). These measurements have been consistently reproduced across multiple independent experiments, with mass variations within ± 200 Da. Although there is a discrepancy between the theoretical and observed molecular weights, the margin of error is approximately 3–6% for these large protein complexes (>500 kDa). We cannot rule the possibility that the recombinant GS1•GYG2 and GS1•GYG1 (Y195F) complexes undergo partial degradation or post-translational modifications isolated from insect cells.

Table. 1 to Reviewer #3

GYG1-GYG2	Theoretical mass (Da)	Measured mass (Da)	% Mass Error
Apo	568262.76	534837.80	-6.25%

GYS1-GYG1(Y195F)	Theoretical mass (Da)	Measured mass (Da)	% Mass Error
Apo	517263.6	534845.90	3.29%

Minor points:

Y-axis values are missing in Fig 2o.

We apologize for this mistake, which has now been corrected.

References 20 and 38 appear to be duplicated.

Nilsson, J. et al. LC-MS/MS characterization of combined glycogenin-1 and glycogenin-2 enzymatic activities reveals their self-glucosylation preferences. *Biochim Biophys Acta* 1844, 398-405 (2014).

We apologize for this error, which has now been corrected.

Reviewer #3

Reviewer #4 (Remarks to the Author)

Weng et al.

In this study Weng and colleagues use a human pluripotent stem cell model to study the role of glycogenins in the regulation of glycogen biosynthesis and cellular glucose metabolism. First the authors use knockout and overexpression models to examine the role of GYG1 and GYG2 in glycogen storage and the regulation of glycogen synthase. Then they undertake protein structure analysis of the complex of both gene products with glycogen synthase to elucidate differences in this interaction between GYG1 and GYG2. They conduct cellular metabolomics analysis in knockout and overexpressing cell lines to demonstrate how the two glycogenins function during challenges to glucose homeostasis, and their influence on oxidative phosphorylation and glycolysis. They use electron microscopy and biochemistry to analyze effects of the knockouts on glycogen storage granules in undifferentiated cells and then in four types of differentiated cell. The overall conclusion is that the two glycogenins function together

in a complex fashion to control glycogen metabolism.

This analysis of the function of the glycogenins in a human cellular model is potentially significant, since the GYG2 gene is primate specific and mouse models are uninformative regarding its role. A major strength of this study is its very thorough and in-depth approach incorporating cellular models, biochemistry, metabolomics, structural biology, and transmission EM. The work is certainly a significant contribution to the detailed knowledge of the function of these two genes in metabolic regulation. I am less convinced that the study has much new to teach us about disease, or that it has “implications for treating glycogen storage diseases and metabolic disease” (from the Abstract).

We thank the reviewer for recognizing the strengths of our study, particularly our comprehensive approach that integrates cellular models, biochemistry, metabolomics, structural biology, and transmission EM. We are pleased that the reviewer views our study as a significant contribution to understanding the roles of these two genes in metabolic regulation.

Regarding the reviewer’s concern about the disease relevance of our findings, we respectfully assert that our study provides key mechanistic insights that may have broader implications for glycogen storage diseases (GSDs) and metabolic disorders. While we acknowledge that further translational research is necessary to establish a direct clinical impact, we believe our findings advance our understanding of the molecular mechanisms underlying glycogen dysregulation, which could inform future therapeutic strategies.

Here the main novelty lies in results with GYG2 since the function of GYG1 has been widely studied and is reasonably well understood. Throughout the study, strong GYG2 phenotypes are only obvious in the context of GYG1 knockouts or overexpression (with one exception, glycogen particle size in ESC), both somewhat artificial conditions. This is not surprising given the data from human genetics: reference 16 concerns a family with a deletion on the X encompassing GYG2 and is cited by the authors to suggest that the deletion has something to do with diabetes, also occurring at a relatively high frequency in this family. However, reference 16 makes clear that the diabetes is not segregating with the deletion, and that further study from the authors of Ref 16 and others has revealed that roughly 1/100 males in the general population bear similar deletions (which does not rule out some role for the glycogenin GYG2 but is consistent with the relatively weak phenotype of GYG2 KO through most of the authors’ analyses).

We thank the reviewer for their detailed assessment and for highlighting these important points. We agree with the reviewer’s observation that the function of GYG1 has been well studied, and we readily appreciate the opportunity to emphasize the novelty of our findings regarding GYG2. We would like to clarify that in our GYG2 knockout (KO) experiments, we also observed a significant increase in glycogen content, providing further evidence that GYG2 plays a

functional role in glycogen metabolism and particle assembly. This finding demonstrates that GYG2's role is not limited to conditions involving GYG1 knockout or overexpression.

Regarding the cited reference (Ref. 17 in the revised manuscript), we acknowledge that the study reported no clear segregation of the GYG2 deletion with diabetes. However, we would like to highlight that their data showed that 8 out of 12 diabetic patients carried a GYG2 deletion, and that 8 out of 11 individuals displaying GYG2 deletion were diagnosed with diabetes. Even though this correlation does not establish causation, it indicates that GYG2 deletion may contribute to metabolic disturbances in certain contexts. Given the complexity of metabolic diseases such as diabetes, it remains possible that GYG2 deletion influences glycogen metabolism under specific conditions, particularly in combination with other genetic or environmental factors.

Our study provides direct evidence that GYG2 plays a distinct and important role in glycogen metabolism and particle assembly. While we acknowledge that additional research is necessary to fully explore its potential contribution to metabolic diseases, we believe our findings offer valuable mechanistic insights that may inform future investigations in this area.

I am also uncertain about the cellular modelling strategy to elucidate disease mechanisms. It is clear from prior work and the authors' Figure 7 that the actions of glycogenins are highly cell type specific. In this regard it might have made more sense to focus the model on differentiated cells like hepatocytes or cardiomyocytes rather than undifferentiated ESC.

We thank the reviewer for raising this important point regarding our cellular modeling strategy. We agree that glycogenin activity is highly cell type-specific, as demonstrated in previous studies (Ref 12; PMID:35690592) and our own data presented in Figure 7. Indeed, differences in GYG1 and GYG2 expression across various cell lineages can significantly influence glycogen metabolism and particle size.

While we acknowledge that differentiated cells such as hepatocytes or cardiomyocytes are physiologically relevant models for studying glycogen metabolism, we believe that our use of undifferentiated ESCs also provides valuable insights. ESCs offer a controlled and versatile system for dissecting the fundamental molecular mechanisms underlying GYG1 and GYG2 function without the added complexity of cell type-specific regulatory factors.

We recognize that ESCs may not fully capture the complexity of specialized cell types. However, we believe our findings provide important mechanistic insights that complement studies using differentiated cell models. These insights may help guide future investigations in more physiologically relevant systems.

It would be helpful if the authors produced a summary figure indicating very clearly what their results show about the role of GYG2 in normal physiology, how this might be perturbed in disease states and in what diseases specifically, and what the implications for therapeutic intervention might be. If they could make this case more strongly, and it is not obvious to me that they have done so thus far, the impact of the study would be increased considerably.

We thank the reviewer for this insightful comment and for suggesting the inclusion of a summary figure to better illustrate the physiological role of GYG2, its perturbations in disease states, and potential therapeutic implications.

To address this point, we have now included a graphical abstract (Page 71) titled "Physiological Relevance of Dynamic Regulation of GYG1 and GYG2 and Their Interactions with GS in Glycogen Metabolism". This figure summarizes our key findings as follows:

1. GYG1 and GYG2 exert distinct roles in glycogen metabolism:
 - a. GYG1 functions as a glycogen primer via its autoglycosylation activity, promoting glycogen synthesis and β -particle formation.
 - b. GYG2, lacking autoglycosylation capability, cannot compensate for GYG1's role. Instead, GYG2 binds to glycogen synthase (GS) and enhances its phosphorylation, thereby reducing GS activity and glycogen synthesis.
2. Regulation of glycogen particle morphology:
 - a. While GYG1 is essential for initiating glycogen synthesis, both GYG1 and GYG2 are required for proper glycogen α -particle formation, highlighting the importance of GYG2 in determining glycogen particle size.
3. Impacts on metabolic balance and disease:
 - a. The dynamic expression of GYG1 and GYG2 in response to metabolic changes is crucial for maintaining glycogen homeostasis. Disruption of this balance, through either GYG1 or GYG2 depletion, alters glycogen content, glycolytic metabolite levels, and glycogen particle size.
 - b. These changes may contribute to metabolic reprogramming and impaired physiological function, which are relevant to glycogen storage diseases (GSDs) and other glycogen-related disorders.

Our findings establish GYG2 as a negative regulator of glycogenesis by modulating GS activity and influencing glycogen particle morphology. We believe that this graphical abstract effectively summarizes these insights and underscores the broader implications of our study in terms of understanding metabolic regulation and potential therapeutic targets.

Specific comments (page numbers refer to PDF):

1. Page 3 introduction. Use consistent terminology (GYG does not refer to a specific human gene, gene names (human) should be capitalized in italics. Mouse *Gyg* gene lower case italics).

We apologize for this error and have now made the necessary corrections, including the specified text on Page 3, line 17, and elsewhere throughout the manuscript to ensure consistency.

2. Page 5 What is phenotype of human GYG2 knockout described in Ref 16 (essentially there is none, see also comments above).

We appreciate the reviewer's careful assessment of this reference. On Page 5 of our manuscript, we cited Irgens et al., 2015 (now Ref. 17 in the revised version) and wrote: "Moreover, glycogen synthesis can proceed in the human liver even in the absence of GYG2, indicating a potential compensatory function of GYG1."

This sentence was intended solely to reflect the metabolic conclusion of that study, which demonstrated that GYG2 is dispensable for hepatic glycogen synthesis and glucagon-stimulated glucose release. This interpretation aligns with the authors' experimental findings, showing that liver glycogen levels and glucagon responses were preserved despite complete loss of GYG2.

In the Discussion section, we also referenced Irgens et al., 2015 to highlight a clinical observation. Among individuals from two families carrying a deletion encompassing the entire GYG2 gene, 8 out of 11 were diagnosed with diabetes. Although the study did not demonstrate clear genotype–phenotype segregation and clinical presentations were heterogeneous, including both insulin-requiring and diet-controlled cases, the enrichment of GYG2 deletion among the diabetic individuals implies a possible association with altered glucose metabolism.

We fully agree that this observation does not establish causality. To better reflect the limitations of the evidence, we have revised the text in the Discussion to clarify that the link between GYG2 deletion and diabetes remains inconclusive. We now emphasize that while glycogen synthesis can still occur in the absence of GYG2, the formation of smaller and less organized glycogen particles may impair metabolic buffering under conditions of stress or fluctuating glucose demand. This altered glycogen architecture may contribute to dysregulated glucose homeostasis in certain contexts.

Revised text in the Discussion (Page 32, lines 18–19 and Page 33, lines 1–16):

“Although the study did not demonstrate a direct genetic link between GYG2 deletion and diabetes, the observed enrichment of GYG2 deletion among the diabetic individuals implies a potential association that may warrant further investigation. Here, we found that GYG2 deficiency leads to the formation of smaller and less structured glycogen particles. Though glycogen synthesis can still occur in the absence of GYG2, these altered particles may be stored less efficiently and contribute to metabolic instability under conditions of increased glucose demand. This scenario is consistent with our mechanistic findings that GYG2 does not effectively prime glycogen synthesis and may act to constrain glycogen synthase activity through conformational modulation.

Taken together, these findings highlight a potential role for GYG2 not only in initiating glycogen formation, but also in regulating particle architecture and energy buffering. Although the contribution of GYG2 deletion to diabetes susceptibility remains inconclusive, our data raise the possibility that loss of GYG2 may reduce metabolic flexibility and impair glucose homeostasis, particularly when additional genetic or environmental stressors are present. Future investigations using tissue-specific models or longitudinal metabolic profiling may help elucidate the broader physiological relevance of GYG2 in energy homeostasis.”

3. Page 7 justify use of undifferentiated hESC as primary model instead of hepatocytes, muscle neurons.

We appreciate the reviewer’s interest in our choice of cellular model. While we fully acknowledge that differentiated cell types such as hepatocytes, cardiomyocytes, neurons, or skeletal muscle are physiologically relevant systems for studying glycogen metabolism, we chose to first examine GYG1 and GYG2 function in undifferentiated human embryonic stem cells (hESCs) to establish a foundational mechanistic framework.

Undifferentiated hESCs offer a simplified and genetically stable background that allows us to dissect the direct molecular effects of GYG1 and GYG2 deficiency without the confounding influence of lineage-specific regulatory networks. This system enabled clear interpretation of autoglycosylation capacity, GS interaction, and particle formation in a controlled context.

Importantly, as described in later sections of the manuscript (Fig. 7 and Extended Data Fig. 7), we also performed studies in differentiated hESC-derived hepatocytes, cardiomyocytes, neurons, and skeletal muscle. These results confirm that the differential effects of GYG1 and GYG2 on glycogen content and particle morphology are preserved in lineage-specific contexts. Thus, we view the use of undifferentiated hESCs and differentiated derivatives as complementary approaches that together strengthen the mechanistic conclusions of this work.

We agree that future investigation in physiologically specialized cells will be important to further elucidate the tissue-specific roles of GYG isoforms, and we are actively planning such studies as the next step of this research.

4. Page 8 minimal GYG2 KO effects in 1d (compared to WT) are difficult to reconcile with a doubling in glycogen content in 1h, as are double KO PAS and glycogen stains in 1g

We thank the reviewer for this insightful comment. We agree that PAS staining is a qualitative method that primarily visualizes glycogen distribution rather than providing precise quantification. Even though our PAS staining results provide a valuable visual confirmation of glycogen accumulation, they lack the sensitivity and accuracy required for detailed measurement of glycogen content.

In our study, we relied on glycogen content assays to obtain quantitative data, which demonstrated a significant increase in glycogen levels in the GYG2 KO condition (Figure 1h). The discrepancy between the minimal changes observed from PAS staining (Figure 1d) and the notable increase in glycogen content may reflect the limitations of PAS staining in detecting subtle differences in glycogen accumulation, particularly in early-stage metabolic changes.

5. Fig 2e-differences in pGS/GS ratio in DKO/OE GYG2 are not convincing. Why doesn't ratio change in G2 KO? How to reconcile double KO effects in 2e with effects on glycogen content in 1h? What is actual variability in control cells of GS and PGS measurement (normalized values only are shown).

We thank the reviewer for these insightful questions. The Western blot data shown in Figure 2d-e were obtained using lysates not treated with α -amylase, and therefore they reflect the non-glycogen-bound (cytosolic) pool of GS and pGS. Under these conditions, both GYG2 KO and DKO cells show low levels of cytosolic GS and pGS, likely due to their redistribution into the glycogen-bound fraction, which is consistent with the increased glycogen content seen in Figure 1h. This outcome explains why the pGS/GS ratio in these genotypes appears unchanged despite elevated glycogen levels.

To address this issue directly, we have now performed additional Western blotting on α -amylase-treated lysates, in which glycogen-associated GS is released. Our results (now included here as Figure 1 to Reviewer 4) show that levels of total GS and pGS are elevated in the GYG2 KO and DKO cells, and that pGS is increased in DKO^{OE:GYG2} cells, indicating inhibition of glycogenesis. This result better reconciles the biochemical data with the observed glycogen phenotypes.

For quantification, GS and pGS were blotted on separate membranes to avoid antibody cross-reactivity. The pGS signal was normalized to β -actin, and the pGS/GS ratio was calculated by direct comparison. We note that Figure 2e was specifically designed to assess the regulatory phosphorylation state of cytosolic GS, which we believe reflects the pool most responsive to GYG2-dependent modulation of GS activation. For this reason, we highlighted the non- α -amylase-treated samples in the main figure, while providing the data on the total GS pool for reference.

Fig. 1 to Reviewer #4 | Increased total GS and pGS signals in GYG2 KO and DKO cell lines are consistent with the elevated glycogen content shown in Figure 1h.

a, Schematic illustrating the localization of glycogen-associated versus cytosolic (non-glycogen-bound) glycogenin proteins under conditions with or without α -amylase treatment. **b**, Western blot analysis of total GS, pGS, and GYG2 levels in α -amylase-treated lysates from WT, GYG1 KO, GYG2 KO, DKO, and DKO^{OE:GYG2} hESCs.

c, Quantification of the ratio of total to free GS levels across genotypes. Increased GS abundance in GYG2 KO and DKO lines corresponds with the elevated glycogen content observed in Figure 1h. Data represent mean \pm SD from three independent experiments.

6. Figure 2m GS/GYG2 activity seems negligible under any condition, results do not justify the conclusion that the complex is involved in any physiological regulatory mechanism.

We thank the reviewer for raising this important point. We fully agree that the GS•GYG2 complex displays minimal enzymatic activity under all tested conditions in Figure 2m. However, we do not interpret this as evidence of a lack of physiological function. Instead, our data support a model in which GYG2 serves as a suppressive or regulatory binding partner, stabilizing GS in an inactive conformation.

This interpretation is supported by our cryo-EM structural analysis, which shows that GS adopts a compact, autoinhibited conformation when bound to GYG2. Moreover, λ PP or G6P treatment increased GS activity in the GS•GYG2 complex by ~4-fold, although absolute activity remained much lower than that of GS•GYG1, indicating that GYG2 may constrain the conformational flexibility of GS and thereby limits its full activation.

Together with the elevated pGS levels and glycogen accumulation observed in cells overexpressing GYG2, our findings provide evidence that GYG2 functions not as an activator, but as a modulator that maintains GS in a less active state, providing a regulatory mechanism to prevent inappropriate glycogen synthesis.

7. How much GYG2 protein is added in 2o-the decline in activity is not large and it would be important to know if these results had any physiological significance.

We thank the reviewer for this insightful question. In the GS activity titration assay (Figure 2o), we used 100 nM of pre-formed GS•GYG1 complex and added recombinant GYG2 protein at increasing concentrations (100, 200, 400, 800, and 1600 nM) to assess the effect of GYG2 on GS activity.

This result reflects a dose-dependent effect of GYG2 on the GS•GYG1 complex, suggesting that excess GYG2 may interfere with GS function, possibly through competitive or structural modulation. We do not claim that these exact concentrations replicate levels in vivo. However, together with other cellular experiments, our study demonstrates that GYG2 overexpression leads to increased pGS and glycogen accumulation, supporting its potential regulatory role.

Taken together, these results support that GYG2 may function as a modulatory factor that influences GS activity through conformational or competitive interactions, particularly in settings where GYG2 expression is elevated or GYG1 is deficient.

8. Figure 2m and Figure 3c is GYG2 inactivating the GS or is it just that it does nothing in the absence of GYG1

We appreciate this important question. Based on our biochemical and structural data, we interpret that GYG2 does not directly inactivate GS, but rather lacks the ability to activate it, in contrast to GYG1.

GYG2 exhibits minimal autoglycosylation activity and therefore cannot initiate glycogen synthesis. When GS interacts with GYG2, the resulting complex appears to maintain GS in an

inactive conformation, as supported by the low enzymatic activity observed under various conditions in Figures 2m and 3c, and by our cryo-EM structural findings.

In this context, GYG2 functions as a non-productive binding partner that neither activates GS nor supports primer formation, and may restrict GS activity by stabilizing a conformation that is less permissive to activation.

9. Page 16-the reference cited here (16) states clearly that the large deletion encompassing GYG2 did not segregate with diabetes in the family under study. The inference that GYG2 has some link to metabolic dysregulation in diabetes is not supported by this study.

We thank the reviewer for this comment and acknowledge that the phrasing in the original version of the manuscript (Page 17 and 18) may have been somewhat misleading. Our intention was not to imply a causal relationship between GYG2 deletion and diabetes. Instead, we aimed to reference the clinical observation from Irgens et al., 2015 (now Ref. 17), in which 8 out of 11 individuals with a GYG2 deletion were reported to have diabetes. Although this finding does not establish genetic linkage or causality, we considered it an interesting context for exploring how GYG2 may participate in glycogen homeostasis and metabolic adaptation.

To avoid any misunderstanding, we have now revised the relevant sentence (Page 18, lines 6-8) to more accurately reflect the observational nature of the clinical data. The revised version now states:

“Although the study did not demonstrate a genetic link between GYG2 deletion and diabetes, the observed co-occurrence in several individuals raises the possibility of a metabolic association that merits further exploration.”

We appreciate the reviewer’s feedback and have made this clarification to ensure that our interpretation remains within the bounds of the original evidence.

10. Figure 5a, b-I am not convinced that changes over time in GYG1 or 2 are significant, neither does it look like the changes in GS/pGS, which are real, are much influenced by the GYG2 knockout. I think the authors have overinterpreted these data.

We thank the reviewer for this thoughtful feedback. The changes in expression of GYG1, GYG2, GS, and pGS shown in Figures 5a and 5b were obtained from three independent biological replicates and were analyzed by means of standard statistical methods. The mean values and standard deviations are shown in the line plots, and statistical significance is indicated where appropriate.

Although we understand that some of the changes in expression may appear modest, we respectfully note that several of them, particularly involving GYG2 and pGS in wild-type cells under conditions of metabolic shift, are statistically significant and consistently reproducible across replicates. These data support a potential regulatory relationship between GYG2 expression and GS phosphorylation in response to changing energy states.

Nevertheless, we appreciate the reviewer's concern and have carefully considered the extent to which these changes in expression are emphasized in our interpretation. Despite the differences not being large in magnitude, we view them as part of a broader regulatory trend that complements our other mechanistic findings.

11. Figure 5c Interaction with GYG2 is minimal under any conditions. OE somewhat enhances changes in co-ip with GYG1.

We thank the reviewer for this important comment. We agree that the interaction between GYG2 and GS/pGS appears weaker than that of GYG1. However, it is important to note that all Co-IP samples were processed and analyzed on the same gel and membrane, with matched blotting and exposure conditions, which enables direct relative comparison of binding strength within the experiment.

Under these standardized conditions, GYG1 consistently exhibited stronger interaction with GS/pGS than GYG2, indicating that GYG2 forms less stable or lower-affinity complexes with GS. Although GYG2's interaction appears modest, it is consistent with its proposed role as a modulatory factor, which contrasts with that of GYG1 that functions as the primary autoglycosylating scaffold in the GS complex.

Despite the relatively weaker interaction, GYG2 overexpression was associated with elevated pGS levels (Fig. 2d,e, 7e,f) and reduced glycogen accumulation (Fig. 1h), indicating functional relevance. Furthermore, Co-IP showed that GYG2 preferentially binds pGS, aligning with the notion that it may stabilize the inactive form of GS.

Taken together, although the GYG2–GS interaction is weaker than that of GYG1–GS, the relative binding difference observed within the same blotting context, combined with consistent biochemical and metabolic outcomes, supports the conclusion that GYG2 plays a regulatory role in modulating GS activity.

12. Figure 5d GYG2 KO enhances glycolysis. This is one of the stronger effects of the KO. But what does this mean?

We thank the reviewer for highlighting this observation. The enhanced glycolysis observed in

GYG2 KO cells (Figure 5d) is indicative of a metabolic shift away from glycogen synthesis toward glucose utilization via glycolysis. We agree that the enhanced glycolysis observed in GYG2 KO cells (Figure 5d) is unlikely to result from increased GS activity. Instead, our findings indicate that GYG2 functions as a negative regulator of glycogen synthase (GS) by binding to GS and promoting its phosphorylation, thereby reducing GS activity.

In the absence of GYG2, this inhibitory interaction is lost, potentially leading to partial restoration of GS activity. However, despite this potential increase in GS activity, the metabolic profile in GYG2 KO cells indicates a stronger shift toward glycolysis rather than enhanced glycogen synthesis. This scenario suggests that glucose utilization in GYG2 KO cells may be redirected to meet immediate energy demands rather than being stored as glycogen.

We have now revised the Discussion to clarify this mechanism and to better explain the observed metabolic shift in GYG2 KO cells.

13. Page 20-again the extension of these findings to diabetes is a stretch.

We thank the reviewer for this important comment and for the consistent feedback throughout the review regarding our references to diabetes. In response to these comments, we have carefully reviewed and revised all relevant sentences in the Results and Discussion sections to ensure that any mention of diabetes or metabolic disease is presented in a cautious and strictly observational manner.

Our intention is not to imply a direct or causal relationship between GYG2 and diabetes, but to provide context for our metabolic findings within the broader field of glucose homeostasis. We have now limited all references to diabetes to descriptive statements about the co-occurrence observed in previous studies, and have removed or rephrased any language that could be interpreted as implying pathophysiological significance.

We appreciate the reviewer's feedback and have made these revisions to ensure that our interpretation remains appropriately focused and aligned with the available evidence.

14. Figure 6f-it is difficult to interpret data on the combined OE of GYG1 and GYG2 that seems to restore production of cauliflower like particles to the DKO line. The situation is artificial; data on the proportion of alpha particles should be provided.

We thank the reviewer for this helpful suggestion. We agree that overexpression of both GYG1 and GYG2 in DKO cells represents an artificial system, but we believe it offers mechanistic insights into the role of these isoforms in glycogen particle architecture.

To quantify the formation of α -particles, we measured the proportion of cauliflower-like glycogen particles under different levels of co-overexpression. In the weakly overexpressing (OE) lines, 0.66% of particles were α -like; in the modestly OE lines, 1.91% adopted this conformation; and in the strongly OE lines they accounted for 5.0%. These values are derived from counts conducted blindly across multiple TEM fields (now included in Extended Data Fig. 6c).

Although this model does not replicate physiological expression levels, the dose-dependent increase in α -particles supports the view that both GYG1 and GYG2 are required for proper higher-order glycogen particle assembly. We have now added these quantitative data to the revised manuscript to strengthen our model interpretation.

15. Page 22-the characterization of the differentiated cells is insufficient.

We thank the reviewer for this important comment. The hESC-derived hepatocytes, cardiomyocytes, neurons, and skeletal muscle cells used in Figure 7 were characterized using both morphological features and well-established lineage-specific markers, including alpha-fetoprotein (AFP) for hepatocytes, TNNT2 for cardiomyocytes, TUJ1 for neurons, and ACTN2 for skeletal muscle cells, as shown in Extended Data Figure 7a of the revised manuscript (previously Extended Data Figure 5a).

To further validate that GYG1 or GYG2 knockout does not affect lineage differentiation, we have now performed additional flow cytometry analysis of these markers across the WT, GYG1 KO, and GYG2 KO lines. These new results demonstrate no significant differences in differentiation efficiency among the genotypes, as presented in Extended Data Figure 7a (right panel) and 7b. This outcome confirms that the observed phenotypes are not due to disrupted lineage specification.

In addition to marker validation, we performed multiple layers of functional and structural characterization, including glycogen content quantification (Fig. 7d), GS phosphorylation (Fig. 7e–f), glycogen particle morphology via TEM (Fig. 7b–c, g–j), and pharmacological inhibition experiments (Fig. 7k–l). These data collectively support the suitability of the differentiated cell types for investigating the cell-type-specific roles of GYG1 and GYG2 in glycogen metabolism.

We hope these clarifications and newly included data address the reviewer's concern.

16. Figure 7 d, e, and f-effects of GYG2 KO are unremarkable.

We thank the reviewer for this comment. We agree that the effects of GYG2 knockout are more

subtle compared to the striking phenotypes observed in the GYG1 knockout lines. However, we believe this difference reflects their distinct biological roles. GYG1 is the primary autoglycosylating glycogenin isoform, whereas GYG2 appears to function more as a modulatory factor that influences GS activity and glycogen particle structure under specific metabolic contexts.

In our differentiated cell models, GYG2 KO led to significant increases in glycogen content in hepatocytes and cardiomyocytes (Fig. 7d), as well as reduced GS phosphorylation and altered GYG1 expression (Fig. 7e–f). These changes, although less dramatic than those seen in GYG1 KO cells, are statistically significant and reproducible across experiments.

Taken together, we interpret these results as supporting a role for GYG2 in fine-tuning glycogen metabolism, rather than serving as a primary initiator of glycogen synthesis. This functional distinction is consistent with our biochemical and structural findings.

17. Figure 7d indicates that the effects of GYG1 KO are very cell type dependent, a fact that questions the use of undifferentiated ES cells as a model for most of the work. Note also that these differentiated cells are almost certainly of fetal phenotype.

We thank the reviewer for this insightful comment. We agree that the effects of GYG1 knockout are cell-type specific, as shown in Figure 7d. Rather than being a limitation, we believe this observation reflects a meaningful aspect of the physiological diversity in glycogen regulation across tissues.

Undifferentiated hESCs were used in the majority of our mechanistic analyses because they provide a genetically stable and controlled background that facilitates dissection of the core molecular interactions between GYG1, GYG2, and GS. This simplified system enabled us to characterize the fundamental features of these interactions without the confounding influence of lineage-specific metabolic programs.

To address the reviewer's concern about the developmental state of differentiated cells, we have extended our analysis to later-stage hepatocyte differentiation. Specifically, in day 50 hepatocytes, we observed glycogen content changes in GYG1 KO cells consistent with those seen in Figure 7d (see Fig. 2 to the reviewer below), indicating that these metabolic effects are maintained even in more mature, post-fetal-like cell states.

We appreciate the reviewer's point and believe that our use of both undifferentiated and differentiated hESC-derived lineages strengthens our overall model interpretation by combining mechanistic clarity with cell-type-specific relevance.

Fig. 2 to the Reviewer #4 | D50 hepatocytes show decreased glycogen content in GYG1 KO cells, and increased glycogen content in GYG2 KO cells.

18. Can the authors demonstrate any consequences of reduction of particle size in the mutant lines?

We thank the reviewer for this important question. Even though our study was not specifically designed to assess the functional consequences of glycogen particle size, several observations indicate that particle architecture may influence metabolic outcomes.

Previous studies have reported that larger α -particles are associated with more stable glycogen storage, whereas smaller β -particles or fragmented forms are more susceptible to enzymatic degradation and thus contribute to unstable glucose homeostasis (PMID:22248338 and PMID:29483195). In our study, the reduction of α -particles and formation of smaller glycogen particles in GYG1 KO, GYG2 KO, and DKO cells was accompanied by changes in total glycogen content (Fig. 1h), altered GS activity and phosphorylation (Fig. 2–3), and shifts in glucose metabolism toward glycolysis (Fig. 5d).

These findings support that the smaller glycogen particles in the mutant lines may be less efficiently stored or more readily mobilized, contributing to metabolic dysregulation under stress or fluctuating energy demands. While direct functional testing of particle size effects remains a future research direction, the observed correlation between altered particle architecture and metabolic changes supports a physiological role for proper glycogen structuring.

Reviewer #1 (Remarks to the Author)

In this revised version of the manuscript, Weng and colleagues have addressed most of the comments suggested by reviewer 1. They have performed new experiments and modified the manuscript following the recommendations, which strengthens the article.

Reviewer #2 (Remarks to the Author)

We thank the authors for their rebuttal. They have addressed almost all of the issues that we identified, and we are overall supportive of publication. However there are still some minor points to be addressed.

The authors have incorporated the liver-specific GS2-GN2 complex, and they have clarified the GS isoform used throughout the experiments. They have justified the use of phosphatase in activity assays in response to the reviewers. They have also clarified the text in the paper to no longer infer that GN2 is causing GS phosphorylation, which was one of our concerns.

They have also addressed our point regarding cross-reactivity of the GS and pGS antibodies in a sufficient manner.

They have removed the comparison between stability of an AlphaFold model and a crystal structure, and have instead compared like-for-like AlphaFold models. I also appreciate the identification of specific residues mediating the stability effect that they describe.

Some minor points to address:

- Fig. 4f is very unclear, the residues described are hard to see, and the labels are obscuring sections of the models. I would suggest a more zoomed-in view with clearly visible and labelled residues, as in Fig. 4g, rather than the full monomer images displayed.

We apologize for the confusion. We have included a zoomed-in view of C-loop and clearly indicated key residues in GYG1 and GYG2 that may contribute to the differences in autoglycosylation activity (New Fig. 4g).

- Why are the organisms shown in Figure. 4e different between the GYG1 and GYG2 models? Both human GN isoforms are shown, but the other orthologs should either compare GN1 and GN2 from the same organisms, or the orthologs could be removed entirely and the comparison could be limited to the human enzymes only. The Drosophila GN1 model is particularly unhelpful, as it does not closely follow the same fold as the human enzyme, and it obscures the rest of the models.

We thank the reviewer for the suggestions. To avoid the confusing, we have removed the structural comparison among different species from the main figures, and retained only the primary protein sequence alignments in the Supplementary Figure 4e. As reviewer suggested, we have also included new Zoomed-in images to emphasize the differences in the C-loop region (New Fig. 4g).

- The inter-subunit interaction that they show in Figure. 4g are convincing, but these residues are also conserved in GN2, so they cannot be used as further evidence that the GN1 dimer is more stable than the GN2 dimer. The observation of the interactions may stand, but the text should be clarified to reflect this.

We thank the reviewer for the suggestions. We have moved original Figure 4g to new Figure 4e, and clarified this point by including a new panel to the supplementary figure (Supplementary Figure 4f) and reorganizing the text in the revised manuscript accordingly (Page 17, line 9-19 and Page 18, line1-3).

- In Fig. 1 GN2 is labelled as 66 kDa, but its predicted molecular weight is 55 kDa. They address this in a response to reviewers by saying that it runs differently on a gel due to electrophoretic mobility effects, but this would not change the actual mass, only the predicted mass range that it runs at on a gel. Unless the sample has been subjected to MS to determine the accurate mass in solution, it should be labelled with its true molecular weight and the clarification can be indicated by a note (perhaps with with an Asterix?).

We thank the reviewer for this helpful suggestion. While the theoretical molecular weight of GYG2 (UniProt ID: O15488-1) is approximately 55 kDa, SDS-PAGE consistently shows GYG2 migrating at ~66 kDa, including in purified recombinant form from insect cells (Fig. 2i). In accordance with the reviewer's recommendation, we have revised Fig. 1 to indicate the theoretical

molecular weight (55 kDa) and included a dagger (†) with a corresponding note in the figure legend (Page59, line12-14). The observed shift in electrophoretic mobility of GYG2 may result from protein intrinsic properties and/or post-translational modifications.

Overall, I believe that the main points of the paper are supported by the evidence provided, and that the work is of good quality. The demonstration of structural data, in vitro assays, and cell biology work is thorough and ultimately convincing. The paper addresses a key question in the field (i.e. the role of the two glycogenin isoforms in glycogen particle morphology and regulation).

Reviewer #4 (Remarks to the Author)

Weng et al. R1

In their revision the authors address most points raised in my first review. They add some additional data to the manuscript and to the rebuttal to support their claims, and they have revised the text to clarify some key points and to add some caveats to their interpretation of the results. The graphic abstract also enhances the clarity of the presentation. Although the data implicating GYG2 in metabolic disease are far from convincing, I take the authors' point that the genetics of diabetes is in general quite complex and that variants in GYG2 could in certain genetic backgrounds and under certain environmental conditions contribute to the phenotype.

There are a few instances in the rebuttal where the authors offer clarification in response to the review but do not modify the text. I think the points below could be addressed in the text of the manuscript to provide additional clarity.

1. Page 8 Figure 1

“In our study, we relied on glycogen content assays to obtain quantitative data, which demonstrated a significant increase in glycogen levels in the GYG2 KO condition (Figure 1h). The discrepancy between the minimal changes observed from PAS staining (Figure 1d) and the notable increase in glycogen content may reflect the limitations of PAS staining in detecting subtle differences in glycogen accumulation, particularly in early-stage metabolic changes.”

We thank the reviewer for this helpful suggestion. We have now incorporated the clarification into the Results section (Page 8, lines 14–18) to address the discrepancy between PAS staining and quantitative glycogen content assays. This addition provides further explanation regarding the sensitivity of PAS staining in detecting modest changes in glycogen levels.

2. Figure 2m and Figure 3c

“In this context, GYG2 functions as a non-productive binding partner that neither activates GS nor supports primer formation and may restrict GS activity by stabilizing a conformation that is less permissive to activation.”

We thank the reviewer for this suggestion. We have now incorporated the proposed summary sentence at the end of the paragraph describing Figures 2m and 3c (Page 13, lines 2-5), to better highlight the dual role of GYG2 as a non-productive binding partner and negative regulator of GS activity.

3. Figure 5d

“Instead, our findings indicate that GYG2 functions as a negative regulator of glycogen synthase (GS) by binding to GS and promoting its phosphorylation, thereby reducing GS activity. In the absence of GYG2, this inhibitory interaction is lost, potentially leading to partial restoration of GS activity. However, despite this potential increase in GS activity, the metabolic profile in GYG2 KO cells indicates a stronger shift toward glycolysis rather than enhanced glycogen synthesis. This scenario suggests that glucose utilization in GYG2 KO cells may be redirected to meet immediate energy demands rather than being stored as glycogen.”

We have now revised the Discussion to clarify this mechanism and to better explain the observed metabolic shift in GYG2 KO cells.

We have now incorporated this explanation into the Discussion section (Page 31, lines 6–12) to clarify the relationship between GYG2 loss, GS activity, and the observed glycolytic shift in GYG2 KO cells.

I am not sure where this is addressed in the revised discussion.

4. Glycogen particles

“Previous studies have reported that larger α -particles are associated with more stable glycogen storage, whereas smaller β -particles or fragmented forms are more susceptible to enzymatic degradation and thus contribute to unstable glucose homeostasis (PMID:22248338 and PMID:29483195). In our study, the

reduction of α -particles and formation of smaller glycogen particles in GYG1 KO, GYG2 KO, and DKO cells was accompanied by changes in total glycogen content (Fig. 1h), altered GS activity and phosphorylation (Fig. 2–3), and shifts in glucose metabolism toward glycolysis (Fig. 5d).”

We thank the reviewer for pointing this out. We have now integrated the explanation regarding glycogen particle size and its metabolic implications into the Discussion section (Page 33, lines 18-19; Page34, line1-6), including citations to the referenced studies (PMID: 22248338; 29483195).